# Brominated lipid probes expose structural asymmetries in constricted membranes

**Frank R. Moss III** [1,2,8,9]**, James Lincoff** [3,4,9]**, Maxwell Tucker** [3,4]**, Arshad Mohammed** [1,5,8]**, Michael Grabe** [3,4] **& Adam Frost** [1,6,7,8]

Lipids in biological membranes are thought to be functionally organized, but few experimental tools can probe nanoscale membrane structure. Using brominated lipids as contrast probes for cryo-EM and a model ESCRT-III membrane-remodeling system composed of human CHMP1B and IST1, we observed leaflet-level and protein-localized structural lipid patterns within highly constricted and thinned membrane nanotubes. These nanotubes differed markedly from protein-free, flat bilayers in leaflet thickness, lipid diffusion rates and lipid compositional and conformational asymmetries. Simulations and cryo-EM imaging of brominated stearoyl-docosahexanenoyl-phosphocholine showed how a pair of phenylalanine residues scored the outer leaflet with a helical hydrophobic defect where polyunsaturated docosahexaenoyl tails accumulated at the bilayer surface. Combining cryo-EM of halogenated lipids with molecular dynamics thus enables new characterizations of the composition and structure of membranes on molecular length scales.

Cells use molecular machines to form and remodel their membrane-defined compartments' compositions, shapes and connections. The regulated activity of these membrane-remodeling machines drives processes such as vesicular traffic and organelle homeostasis. However, the precise mechanisms by which proteins generate mechanical force to catalyze membrane fission, fusion and shape changes remain elusive. Further, the structural evolution of lipids and lipid bilayers during these processes are challenging to study, as is quantifying the energetic contributions of individual lipid species to membrane remodeling. A better understanding of membrane and lipid structure during remodeling is crucial to understanding both the mechanisms of membrane remodeling and, more broadly, the interactions between proteins and membranes.

Molecular-scale insights into the lipid–leaflet, lipid–lipid and lipid–protein dynamics that generate extreme membrane curvature could clarify the mechanisms of membrane remodeling. A thorough understanding of membrane mechanics will account for lipid asymmetry and flip/flop[1,2], spontaneous lipid curvature[1,3–6], bending rigidity[2,5,7], line tensions[8] and protein-generated forces including amino acid insertions[5,9,10], lateral pressure from protein crowding[11], shearing forces[7] and lipid–protein 'friction'[12]. The relative contributions of each of these processes to membrane remodeling generally, and to membrane constriction in the present case, are challenging to measure experimentally or explain in structural terms. We need new measures of how membrane structure, composition and protein-generated forces influence membrane properties in vitro and in vivo.

So far, cryo-EM reconstructions of membrane-bound proteins have generated insights into the machines that shape membranes, including dynamin family proteins[13–17], OPA1 (ref. [18]), LPOR[19], SNAREs[20], BAR domain containing proteins[21] and ESCRTs[22]. However, these studies have generally failed to reveal molecular-level information about the membrane itself because most lipids display nearly indistinguishable electron scattering[23,24]. Fluid bilayers, moreover, are generally thought to lack a structured pattern at the nanoscale that is recoverable by

[1]Department of Biochemistry and Biophysics, University of California San Francisco, San Francisco, CA, USA. [2]SLAC National Accelerator Laboratory, Menlo Park, CA, USA. [3]Department of Pharmaceutical Chemistry, University of California San Francisco, San Francisco, CA, USA. [4]Cardiovascular Research Institute, University of California San Francisco (UCSF), San Francisco, CA, USA. [5]University of California Berkeley, Berkeley, CA, USA. [6]Chan Zuckerberg Biohub, San Francisco, CA, USA. [7]Department of Biochemistry, University of Utah, Salt Lake City, UT, USA. [8]Present address: Altos Labs, Redwood City, CA, USA. [9]These authors contributed equally: Frank R. Moss III, James Lincoff. ✉e-mail: michael.grabe@ucsf.edu; afrost@altoslabs.com

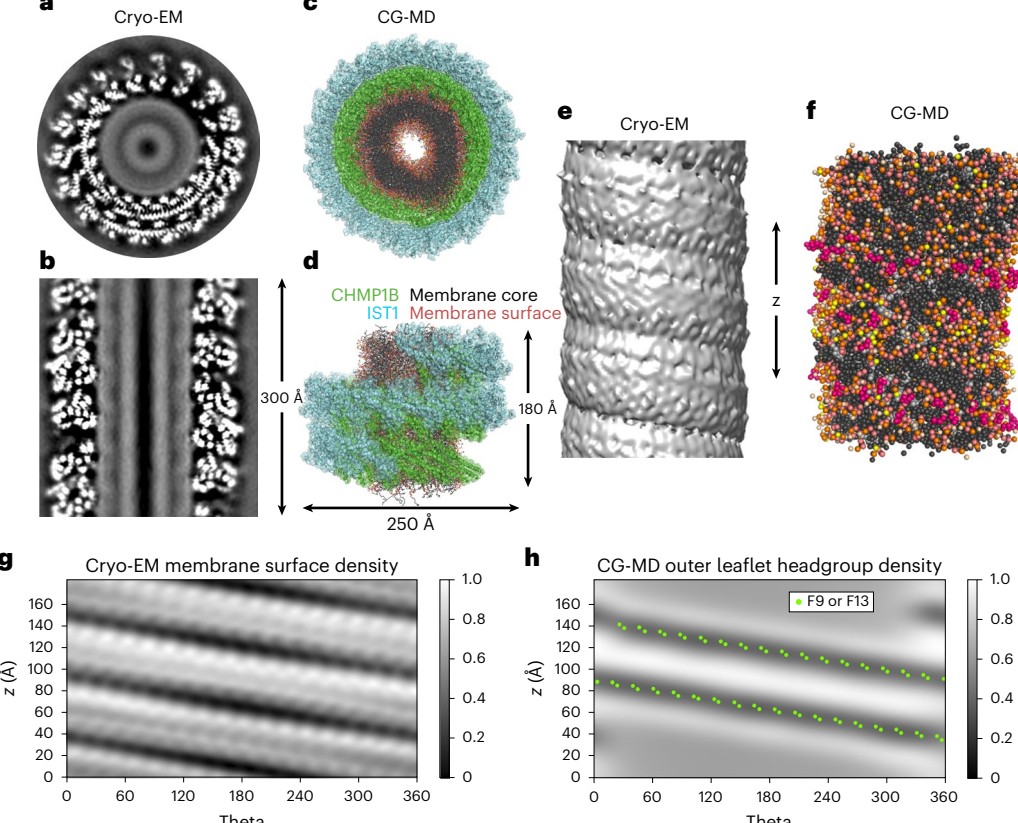

**Fig. 1 | Cryo-EM and CG-MD structures of membrane deformations induced by CHMP1B and IST1. a,b,** Gray-scale top-down (**a**) and side views (**b**) of central slices of the cryo-EM density for the CHMP1B–IST1 copolymer bound to a cylindrical lipid bilayer comprising the lipids shown in composition no. 1, Table 1. **c,d,** Top-down (**c**) and side views (**d**) of an equilibrated tubule within the protein coat from CG-MD simulations, in all-atom representation. CHMP1B in green, IST1 in cyan, cholesterol in light gray, lipid tails in dark gray, phospholipid glycerol moieties in salmon, phospholipid phosphates in orange, SDPC headgroups in beige, POPS headgroups in yellow and PIP₂ headgroups in pink. See PDB 6TZ5 for

the protein structure. **e,** Side view of the lipid bilayer nanotube's cryo-EM density with protein masked. **f,** Side view of the simulated CG-MD membrane tubule with protein hidden shows two bands where headgroups are displaced, and lipid tails are exposed. Color scheme is the same as in **c** and **d**. **g,** EM density of the outer leaflet of the bilayer visualized in cylindrical coordinates. The periodic footprint of CHMP1B is apparent at this depth. **h,** Normalized two-dimensional headgroup density of outer leaflet phospholipids from CG-MD simulations (headgroup exclusion zone appears as dark low-density bands) with F9 and F13 side chain locations (green dots). Data for graphs in **g** and **h** are available as source data.

image averaging. Therefore, it is typically impossible to distinguish individual lipid species, even approximately, except in high-resolution structures where isolated, static lipids are resolved bound to transmembrane proteins[25–28]. To overcome this shortcoming, here we describe halogen-labeled lipid probes as contrast-enhancing agents for cryo-EM. We use a model system consisting of human ESCRT-III proteins CHMP1B and IST1 to remodel vesicles containing lipid probes into lipid nanotubes with extremely high curvature[22,29,30]. Using labeled lipids, ESCRT-III proteins, cryo-EM and molecular dynamics simulations, we characterize how ensembles of lipid-protein interactions, lipid conformational changes and the resulting stabilization of strong membrane asymmetries drive membrane constriction and thinning. Leaflet and lipid shape change together with specific interactions between lipids and the ESCRT-III proteins, stabilize membrane curvature in this snapshot of a membrane on the verge of scission.

## Results

### ESCRT-III protein CHMP1B α1 induces a furrow in the bilayer

When exposed to model membranes in vitro, ESCRT-III proteins can shape membranes into structures with zero[31,32], positive[22,29,33–36] or negative curvature[31,32,37–39]. We recently demonstrated how a pair of human ESCRT-III proteins, CHMP1B and IST1, assemble into helical filaments that act in sequence to remodel membranes into high-curvature nanotubes with an inner diameter of only roughly

40 Å—nearly to the point of fission[22,40]. This system provides an opportunity for probing how peripheral membrane proteins engage distinct lipid species to pattern membrane properties more generally. Additionally, this highly constricted state may be a metastable or intermediate state on pathway to membrane fission and therefore may illuminate the mechanical and molecular properties underlying membrane fission reactions. Previously, we observed that helix α1 of CHMP1B appeared to 'dimple' the outer surface of highly constricted membrane nanotubes, but the properties of this deformation were poorly defined (Fig. 1a,b,e)[22]. To explore the membrane's structure in more detail, we performed coarse-grained-molecular dynamics (CG-MD) simulations of the membrane tubule in the presence of the CHMP1B–IST1 copolymer using the Martini 2.2 CG force field (Fig. 1c,d and Extended Data Fig. 1). We used the previously optimized mixture of stearoyl-docosahexanenoyl-phosphatidylcholine (SDPC), palmitoyl-oleoyl-phosphatidylserine (POPS), di-oleoyl-phosphatidylinositol-3,5-bisphosphate] (PIP₂) and cholesterol (CHOL) for the simulation and subsequent cryo-EM experiments (see Supplementary Fig. 1 for chemical structures)[22]. Simulations were started with a spontaneous tubule assembly MD phase with randomized initial lipid positions between replicates, followed by an extended leaflet equilibration during which lipids flip-flop at artificially induced membrane pores to arrive at consistent inner and outer leaflet compositions across replicates before production data collection (see Methods and

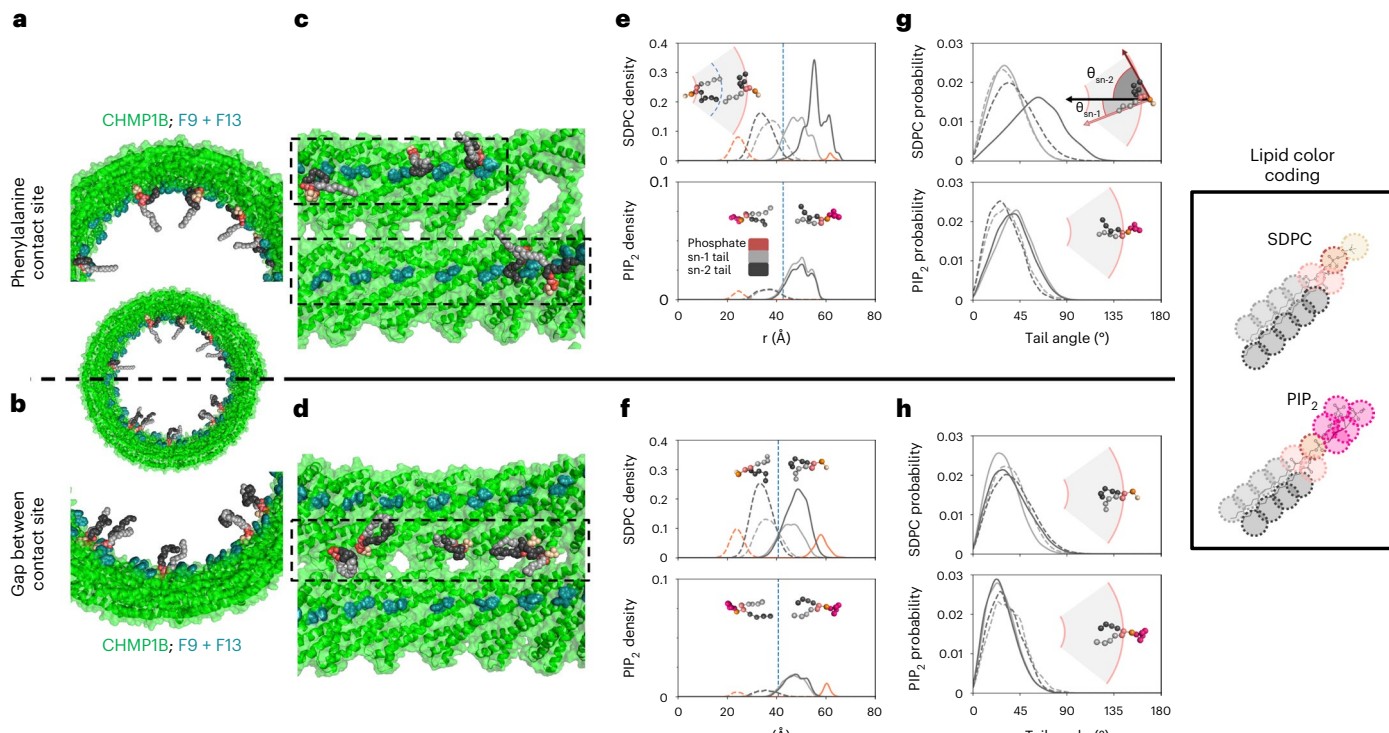

**Fig. 2 | Analysis of lipid conformation and geometry. a**, Representative SDPC molecules from a CG-MD snapshot backmapped to all-atom representation with sn-2 tail (black) interacting with F9 and F13 (teal) of CHMP1B (green), while the sn-1 tail (gray) is normally oriented. SDPC headgroup carbons are shown in beige and glycerols in salmon. Full color coding in legend at right. **b**, Representative backmapped SDPC molecules from a CG-MD snapshot in the gap between CHMP1B α1 helices. **c**, Side view of lipids in **a**, looking from the tubule center toward CHMP1B. Dashed boxes highlight the spiral of F9 and F13. **d**, Side view of lipids in **b**. Dashed box highlights the gap zone between stripes of F9 and F13. **e**, Radial densities of lipid components at bilayer contact sites for SDPC (top) and PIP$_2$ (bottom). Inner leaflet densities in dashed lines, outer leaflet densities in solid lines. Color coding highlights phosphate bead (orange), all sn-1 tail beads (gray) and all sn-2 tail beads (black). Vertical dashed lines denote the local bilayer midplane. **f**, Radial densities of lipid components at gaps for SDPC (top) and PIP$_2$ (bottom). Note the different y axis scale for PIP$_2$ in **e** and **f**. **g**, Probability density of lipid tail angles to the bilayer normal for SDPC (top) and PIP$_2$ (bottom) at the Phe contact site. Black lines are sn-2 tails, gray lines are sn-1 tails, solid lines are outer leaflet and dashed lines are inner leaflet. **h**, Probability density of lipid tail angles to the bilayer normal at the gaps. In **e**–**h**, example CG conformations use the same color scheme as **a**–**d**, with PIP$_2$ headgroups in pink to contrast with SDPC headgroups (beige). Data for graphs in **e**–**h** are available as source data.

Supplementary Information for full details). The simulations revealed CHMP1B side chains sterically displacing phospholipid headgroups and exposing the hydrophobic core of the membrane in an apparent 'scoring' of the surface—rather than an elastic dimpling of a continuous headgroup surface as we previously thought[22] (Fig. 1f and Supplementary Fig. 2). We note that unlike the cryo-EM reconstructions, there is no symmetry applied to the bilayer during the simulations and the tubule fluctuates. Deviations from an ideal cylindrical shape are most obvious at the upper and lower boundary, where the imperfect periodicity of the protein coat results in an exposed membrane. The morphology of the bilayer tube, especially in this region, is sensitive to the number of initial lipids (Extended Data Fig. 1).

To aid visualization of the headgroup exclusion zone, we transformed the protein and lipid positions into cylindrical coordinates and plotted the mean density of headgroups in the outer leaflet (Fig. 1h). The headgroup-excluding furrow aligned with the bulky hydrophobic residues F9 and F13, which protrude from helix α1 on CHMP1B (green dots in Fig. 1h). The bilayer appeared unperturbed away from F9 and F13. Notably, these two residues are the only hydrophobic residues that face the bilayer, with the rest of the bilayer-interacting residues containing cationic side chains. Additionally, given the orientation of CHMP1B helix α1 relative to the curved tubule surface, the pair of phenylalanines are the only residues that protrude into the bilayer (Supplementary Fig. 3a). Despite the 23 Å distance between F9 and F13 pairs on adjacent CHMP1B subunits, the furrow forms a continuous stripe of excluded headgroups in the membrane outer leaflet in simulation

(Fig. 1f,h). This aspect of the simulated structures is in agreement with the experimental structure determined by cryo-EM (Fig. 1e,g), with the EM-derived density map revealing a furrow of low Coulombic potential aligned with the stripe of F9 and F13 positions along the CHMP1B helical filament. Together, the EM maps and CG-MD results indicate that the observed deformation constitutes a stable lipid packing defect due to headgroup displacement by these hydrophobic amino acids of CHMP1B and concomitant exposure of the lipid tails underneath (Fig. 1g,h). Line tension associated with the hydrophobic defect may energetically drive the defects to coalesce into a continuous, helical stripe. Because such an asymmetry in the membrane should alter its mechanical properties, influencing the energetic barriers associated with curvature generation or fission, we sought to further characterize its structure.

## Specific lipids enrich at contact sites and change shape

We next examined the molecular structure and dynamics of the lipids throughout the simulated lipid bilayer. Specifically, we analyzed the radii of each of the lipid beads and the angles of each of the lipid tails with respect to the bilayer normal. This analysis revealed pronounced asymmetries in lipid shape correlated with leaflet localization and proximity to the CHMP1B F9, F13 contact site (Fig. 2a–d). The lipid SDPC, which has a polyunsaturated (22:6) tail at the sn-2 position and a fully saturated (18:0) tail at sn-1, showed the strongest asymmetries. While the saturated sn-1 tail typically remained aligned with the membrane normal throughout both leaflets (Fig. 2a–d), the polyunsaturated sn-2 tail of outer leaflet lipids bent backward and radially outward toward

**Table 1 | Lipid mixtures used in membrane-remodeling assays**

| Number | Composition | Components |
|--------|-------------|------------|
| 1 | 58.2:18:17.5:6.3 | SDPC:CHOL:POPS:PIP$_2$ |
| 2 | 62.1:18.7:19.2:0 | SDPC:CHOL:POPS:PIP$_2$ |
| 3 | 58.2:18:17.5:6.3 | POPC:CHOL:POPS:PIP$_2$ |
| 4 | 71:0:21.3:7.7 | SDPC:CHOL:POPS:PIP$_2$ |
| 5 | 63.9:10:19.2:6.9 | SDPC:CHOL:POPS:PIP$_2$ |
| 6 | 56.8:20:17.1:6.1 | SDPC:CHOL:POPS:PIP$_2$ |
| 7 | 53.2:30:16:5.8 | SDPC:CHOL:POPS:PIP$_2$ |
| 8 | 42.6:40:12.8:4.6 | SDPC:CHOL:POPS:PIP$_2$ |
| 9 | 29.1:50:8.8:3.1 | SDPC:CHOL:POPS:PIP$_2$ |
| 10 | 58.2:18:17.5:6.3 | SDPC-Br:CHOL:POPS:PIP$_2$ |
| 11 | 58.2:18:17.5:6.3 | SDPC:CHOL-Br:POPS:PIP$_2$ |
| 12 | 58.2:18:17.5:6.3 | SDPC:CHOL:POPS-Br:PIP$_2$ |
| 13 | 58.2:18:17.5:6.3 | SDPC:CHOL:POPS:PIP$_2$-Br |
| 14 | 58.2:18:17.5:6.3 | SDPC:CHOL-I:POPS:PIP$_2$ |
| 15 | 29.1:29.1:18:17.5:6.3 | SDPC-Br:SDPC:CHOL:POPS:PIP$_2$ |
| 16 | 14.6:43.6:18:17.5:6.3 | SDPC-Br:SDPC:CHOL:POPS:PIP$_2$ |
| 17 | 26:22:32:20 | SDPC:CHOL:POPS:PIP$_2$ |

All quantities are mole percentages.

the headgroup region at the F9, F13 contact site (Fig. 2a,c). This shape transformation partially filled the space vacated by the displaced headgroups and enabled the tail to interact directly with the hydrophobic side chains of F9 and F13. Away from the exclusion zone, the polyunsaturated tail adopted a more typical conformations, largely oriented along the membrane normal (Fig. 2b,d).

To quantify lipid conformation, we calculated the lipid tilt angles and radial densities of the phosphate and tail beads of all phospholipids adjacent to and in between the Phe contact sites for both leaflets (Fig. 2e–h and Extended Data Fig. 2). We defined the 'Phe contact site' as the 30° wedge centered through the angular position of F9 and F13, while the 'gaps between the contact sites' were defined as the 30° wedge centered 180° across from the contact sites (Supplementary Fig. 3). Consistent with our anecdotal observations of tail 'backflipping', the sn-2 tail of SDPC was highly deformed in the outer leaflet at the Phe contact sites with its mean location shifted radially outward and a substantial density beyond both the bilayer phosphate peak, typically the membrane's outer surface, and the Phe side chains (Fig. 2e, $r = 60$ Å). On the other hand, in the inner leaflet, both tails are largely unperturbed throughout (Fig. 2e,f). The other lipid tails showed more minor degrees of deformation at the Phe contact sites and normal bilayer behavior in the gaps between furrows.

The tilt angle analysis revealed corresponding asymmetries (Fig. 2g,h). In both leaflets, SDPC tails adopted typical values of roughly 30° with respect to the bilayer normal—except for the polyunsaturated sn-2 tail at the furrows, which displayed pronounced tilting with a mean value of 60° and some extreme conformations greater than 90° where the tail bead resided at the membrane surface beyond the radial position of F9 and F13. The other two phospholipid species, POPS and PIP$_2$, which have both saturated and monounsaturated lipid tails, did not exhibit a similarly strong location-dependent perturbation (Extended Data Fig. 2a,b).

These observations are consistent with the notion that polyunsaturated lipid tails enable membrane shape plasticity under curvature stress, which previous work has suggested may be a general principle of bilayer deformation[3,22,41–45]. Indeed, the polyunsaturated tail of SDPC

is known to be highly flexible. This property could facilitate membrane bending if the lipid changes its mean shape in response to curvature stress[46–49]. Consistent with this hypothesis, replacing SDPC with monounsaturated palmitoyl-oleoyl-phosphatidylcholine (POPC) precluded high-curvature tubule formation but did not preclude membrane binding (Extended Data Fig. 3). Specifically, CHMP1B and IST1 were unable to constrict POPC-containing liposomes (lipid composition 3) to high curvature, as assessed by negative stain transmission electron microscopy and cryo-EM (Extended Data Fig. 3).

We next sought direct experimental validation of SDPC's localization in the bilayer and whether it undergoes such dramatic shape transformations. To detect individual lipid species using cryo-EM, we synthesized analogs of each lipid with bromine atoms along the unsaturated lipid tail lengths. Due to their more massive nuclei, bromine atoms generate contrast through enhanced electron scattering. The brominated and unbrominated lipids displayed similar bilayer phase behavior and Langmuir monolayer properties, indicating comparable lipid packing and fluidity (Extended Data Fig. 4h,i). Additionally, in bilayers, SDPC-Br and POPS-Br are fluid at room temperature and macroscopically homogeneous, consistent with previous studies[50,51]. CHMP1B and IST1 remodeled brominated vesicles into narrow nanotubes yielding protein structures indistinguishable at roughly 3 Å resolution from protein-bound nanotubes prepared with native lipids (Fig. 3, Extended Data Fig. 3 and Supplementary Fig. 4). Together, these results indicate that halogenation does not meaningfully perturb the properties of these lipids or the bilayers formed from them. We collected high-resolution cryo-EM data and achieved high-resolution reconstructions with vesicles in which each lipid was replaced by its halogenated analog (Table 1, lipid compositions 10–14) or different fractions of SDPC were brominated (Table 1, lipid compositions 10, 15, 16).

To quantitively compare the reconstructions with brominated and unbrominated lipids, the pixel value distributions were normalized to the CHMP1B intensity from radial averages. We first compared horizontal and vertical slices through each filament and observed differences in the intensities of the lipid bilayer leaflets with brominated lipids. In particular, the SDPC-Br reconstruction showed intense spots of enhanced density adjacent to CHMP1B F9 and F13, consistent with SDPC-Br enrichment at the site (Fig. 3c) and corroborating the CG-MD simulation data showing that lipid tails reached the surface of the bilayer. The other brominated lipids, which are monounsaturated, also accumulated at the membrane–protein contact site, but to a smaller extent. Further, we resolved differences in the positions of different lipid species relative to the Phe contact sites, with the brominated SDPC tail in closer proximity to F9 and F13 both radially and axially than brominated PIP$_2$ and POPS tails (Fig. 3e,f). These last two anionic lipids more closely approached K16 and R20, two cationic residues previously identified as critical for CHMP1B membrane binding to membranes[22].

The altered SDPC tail conformation where the bilayer contacts CHMP1B residues F9 and F13 suggests at least two hypotheses. First, lipid headgroup displacement by F9 and F13 creates lipid packing defects and curvature stress that is relieved when lipid tails fill in the packing defects. Second, hydrophobic interactions drive the lipid tails to interact with the nonpolar Phe side chain. These ideas, the first focused on steric packing and the second focused on chemical interactions between lipids and amino acids, are not mutually exclusive. We generated a series of CHMP1B double mutants, F9X + F13X (where X = A, E or L), to test the relative contributions of each possibility.

All of these CHMP1B double mutants still bound to and remodeled native and brominated lipid bilayers, forming membrane-bound copolymers with IST1 that were indistinguishable from the wild-type (WT) CHMP1B/IST1 copolymer in helical symmetry parameters (Fig. 4). F9 and F13 substitution by small nonpolar A side chains eliminated SDPC-Br enrichment at CHMP1B contact sites (in lipid composition 10).

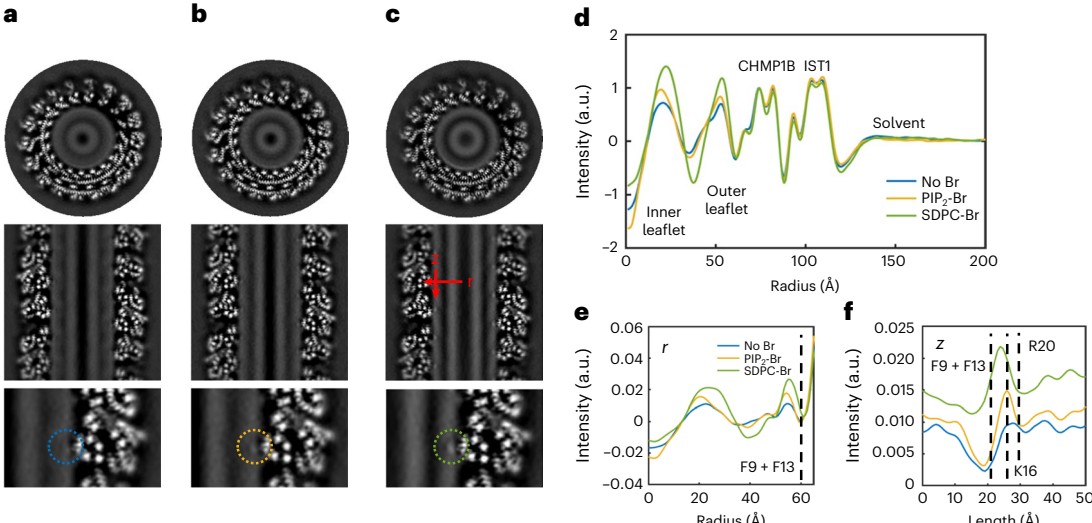

**Fig. 3 | Cryo-EM reconstructions of unlabeled and lipid-brominated membrane-bound CHMP1B/IST1 filaments. a–c**, The top panel shows top-down views of central slices of the reconstructions, the middle panel shows side views of central slices of the reconstructions and the bottom panel shows zoomed-in views of the region where F9 and F13 of CHMP1B α1 contact the membrane: no Br (**a**), PIP2-Br (**b**) and SDPC-Br (**c**). Dashed circles highlight the protein-membrane-contact site. **d**, Radial averages (around the filament axis) of

the reconstructions in **a**–**c** show extra intensity from the bromine labels is located in the lipid bilayer, while the protein structure remains unaffected. See Extended Data Fig. 8a for radial profiles of all brominated lipids mixtures. **e**, Line profile along the filament radius at Phe contact site ($r$, **c** middle panel). **f**, Line profile along the filament axis centered around Phe contact site ($z$, **c** middle panel). The approximate positions of relevant CHMP1B residues along the line scans are indicated by dashed lines. Data for graphs in **d**–**f** are available as source data.

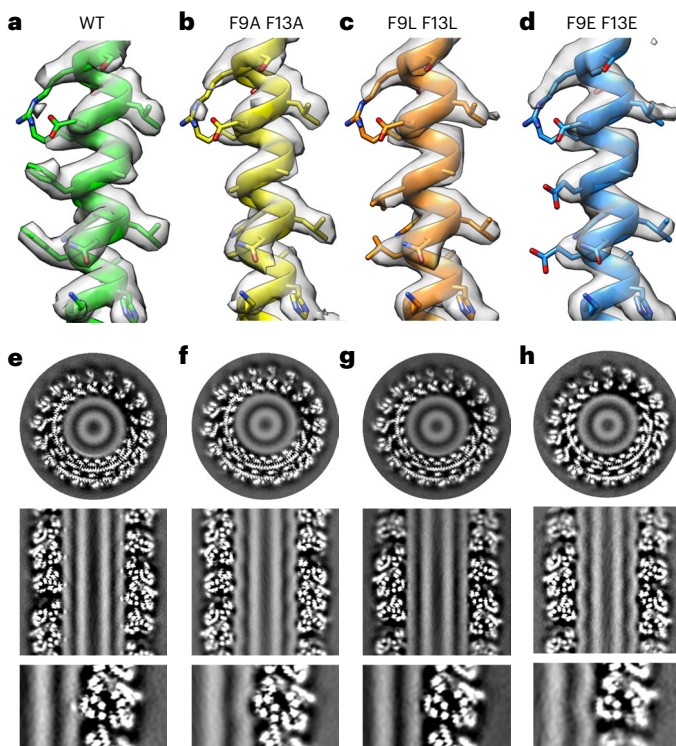

**Fig. 4 | Cryo-EM of CHMP1B F9 and F13 double mutants with SDPC-Br (lipid composition 10). a–d**, A section of CHMP1B helix α1 atomic model (PDB 6TZ5) and cryo-EM density (gray) for WT CHMP1B (**a**) and double mutant proteins, CHMP1B F9A/F13A (**b**), CHMP1B F9L/F13L (**c**) and CHMP1B F9E/F13E (**d**). **e–h**, Horizontal (top) and vertical (middle) slices through the cryo-EM densities for the CHMP1B WT (**e**) and CHMP1B double mutant proteins, CHMP1B F9A/F13A (**f**), CHMP1B F9L/F13L (**g**) and CHMP1B F9E/F13E (**h**). The bottom panels show enlarged images of the membrane-contact sites for each sample. Small and polar side chains induce elastic deformations in the bilayer, while hydrophobic side chains induce hydrophobic defects.

SDPC also failed to concentrate in the F to E double mutant. In both the F to A and the F to E double mutants, the protein appeared to induce an elastic deformation of the bilayer without opening an appreciable hydrophobic defect. The observed deformation in the outer membrane was measurably deeper into the bilayer than for WT CHMP1B and did not induce the helical stripe of low Coulombic potential observed in the WT protein. Simulations of the F to A double mutant also showed reduced SDPC tail enrichment and backflipping, as well as increased headgroup density shifted radially inward, at CHMP1B contact sites, consistent with an elastic deformation (Extended Data Fig. 5). Additionally, SDPC sn-2 tails were less perturbed than with WT protein in simulation. By contrast, in the cryo-EM reconstruction, SDPC-Br was observed to accumulate at contact sites with large hydrophobic Leu side chains to a similar but lesser degree than the WT protein, indicating the formation of a hydrophobic defect. Finally, removal of the protein coat in simulations resulted in dramatic reduction of SDPC tail backflipping and recovered mostly normal bilayer structure and lipid conformation (Extended Data Fig. 6). Together, these observations are consistent with chemical interactions between greasy side chains and polyunsaturated tails stabilizing the backflipped lipid tail conformations. On the other hand, the lack of SDPC sn-2 tail enrichment at the highly curved elastic deformations in the F to A and F to E mutants suggests that curvature stress at the contact site does not play a notable role in SDPC accumulation at WT CHMP1B-bilayer contact sites. These results all indicate that hydrophobic interactions, rather than spontaneous curvature, drive local accumulation of polyunsaturated lipid tails at CHMP1B F9 and F13.

## Constriction leads to anisotropic inner leaflet thinning
Previous work has predicted that a highly curved bilayer's inner, concave leaflet would be thinner than the outer, convex leaflet based on atomistic simulations[52]. We experimentally measured leaflet thicknesses to test this hypothesis by examining the Coulombic potential (electron scattering) profile from the unbrominated sample and fitting the profile with three Gaussian functions using least-squares[53]. As shown in Fig. 5a,c, the inner concave leaflet is 3.6 Å thinner than the

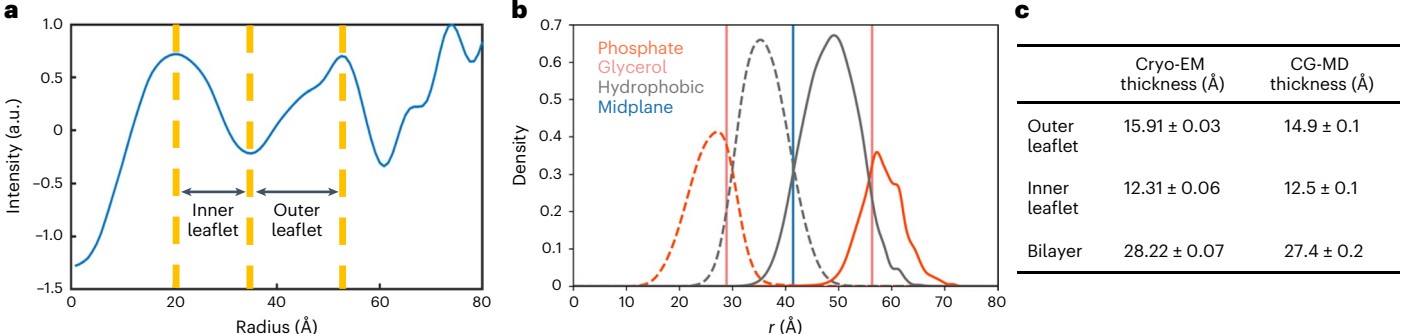

**Fig. 5 | Bilayer and leaflet thicknesses of a membrane nanotube from simulation and experiment. a**, Radial profiles from cryo-EM reconstructions provide estimates of inner and outer leaflet thickness values by computing distances from each leaflet peak to the aliphatic trough. **b**, Radial densities of hydrophobic (gray curves) and hydrophilic (orange curves) membrane components for inner leaflet (dashed lines) and outer leaflet (solid lines) in simulation. Vertical lines denote the position of the midplane (blue) and mean glycerol inner and outer leaflet planes (pink) for calculating leaflet thicknesses. **c**, Summary of calculated leaflet and bilayer thicknesses from cryo-EM and CG-MD. Uncertainty is reported as the standard deviation between independent measurements from each half map for cryo-EM, and as the standard deviation between ten simulation replicates for CG-MD. Data for graphs in **a** and **b** are available as source data.

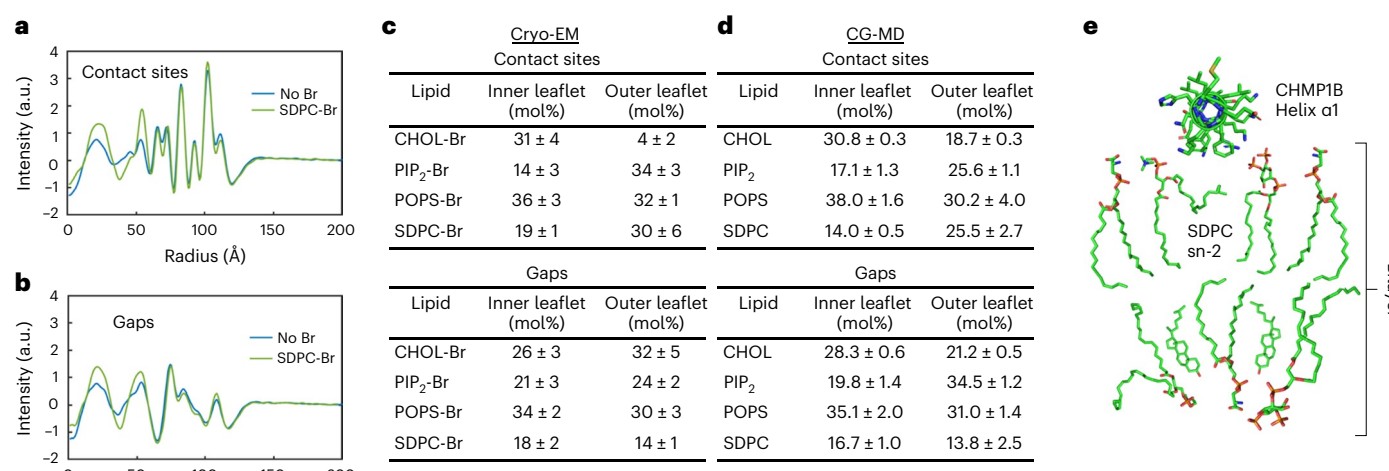

**Fig. 6 | Analysis of lipid bilayer composition at membrane–protein contact sites and between contact sites from simulation and experiment. a,b**, Radial profile of cryo-EM density (**a**) at CHMP1B contact sites and (**b**) between contact sites for the unbrominated and SDPC-Br samples. See Extended Data Fig. 4c,d for all profiles. **c**, The molar composition of each leaflet at membrane–protein contact sites (top) and at the gaps (bottom). Uncertainty was calculated as the standard deviation from independent calculations from each of the two half maps. **d**, The corresponding compositions for these two sections from CG-MD are shown for the contact sites (top) and at the gaps (bottom) from simulations using the enriched lipid composition. Uncertainty was calculated as the standard deviation between three replicates of CG-MD trajectories. **e**, Cartoon model of bilayer structure and composition at CHMP1B contact sites. Data for graphs in **a** and **b** are available as source data.

outer convex leaflet, in agreement with Yesylevskyy et al.[52] and our own CG-MD simulations of the same system (Fig. 5b,c).

We believe that differences in per-leaflet lipid tail angles seen in simulation may explain the different leaflet thicknesses. In the WT CHMP1B–IST1 with lipid composition 1 simulations, inner leaflet lipid tails sample higher tilt angles than those in the outer leaflet except at the F9, F13 contact site (Fig. 2g,h and Extended Data Fig. 2b), where there is extreme outer leaflet backflipping. Throughout the rest of the bilayer, outer leaflet lipids behave like those in the gap zone (Fig. 2h) and sample lower tilt angles. In contrast, to fill the greater volume nearer the bilayer center versus at the inner leaflet headgroups, inner leaflet lipid tails sample higher tilt angles, effectively shortening those lipids in the radial coordinate and leading to the observed inner leaflet thinning. This effect is also present in simulations with the F9A, F13A mutant (Extended Data Fig. 5d,e) and to a lesser extent in simulation

of a protein-free tubule (Extended Data Fig. 6e,f), although that tubule equilibrates to markedly different leaflet compositions than the protein-bound tubules (Extended Data Fig. 6a,c). The increased disorder in the inner leaflet that accompanies higher tilt angles and thinning may facilitate tubule scission by enabling formation of a hemifusion stalk across the tubule lumen.

### The constricted bilayer is compositionally heterogenous

Both MD simulations and cryo-EM reconstructions suggested the bilayer composition is not homogeneous, with differences at the CHMP1B F9, F13 contact site versus the gap between the contact sites. By integrating the areas under the radial curve for each leaflet from bromine-labeled reconstructions normalized to the unlabeled reconstruction, we estimated the composition of each leaflet in these two regions (Fig. 6a,b and Extended Data Fig. 4a–d). As shown in Fig. 6c,

the tubule leaflet composition seems to be different from the starting bulk values (Table 1, composition 1). Anionic lipids are enriched in the filament at the expense of SDPC, likely due to the highly cationic luminal surface of the protein coat (Extended Data Fig. 3a). We hypothesize that such large lipid enrichments can occur because most filaments contain an unremodeled vesicle protruding from the end that can act as a lipid reservoir (for example, Extended Data Fig. 3a). There may also be compositional variation in the initial vesicle population and consequently bias in which vesicles are remodeled and therefore contribute to our cryo-EM reconstructions. We repeated the CG-MD simulations with the composition estimated from cryo-EM (Table 1, lipid composition 17), and this updated simulation displayed analogous asymmetric leaflet thinning and outer leaflet headgroup exclusion displayed by the CG-MD simulations with the initial lipid composition (Extended Data Fig. 7, Supplementary Information Table 8).

Both simulation and experiment indicated that cholesterol was enriched asymmetrically in the inner leaflet versus the outer leaflet at bilayer-protein contact sites. We confirmed the behavior of cholesterol by solving cryo-EM structures with different concentrations of unbrominated cholesterol in the initial lipid vesicles (Table 1, lipid compositions 4–9). We took advantage of the fact that cholesterol has a lower Coulombic potential than phospholipids. As the concentration of cholesterol in the lipid mixtures increased, the inner leaflet decreased in intensity relative to the outer leaflet. This observation is also consistent with the cholesterol enrichment in the inner leaflet seen in simulation (Extended Data Fig. 8d,e).

We also estimated that $PIP_2$ was highly enriched not only in the outer leaflet compared with the inner leaflet, but also in the membrane nanotube compared with the initial bulk composition of the vesicle. In light of this, we next investigated the importance of $PIP_2$ for tubule formation by carrying out membrane-remodeling assays with vesicles containing no $PIP_2$ (Table 1, lipid composition 2). In this case, CHMP1B is unable to stably bind to and remodel vesicles, suggesting that electrostatic interactions are indispensable for its functions (Extended Data Fig. 3i,m). Similarly, K16E or R20E mutations on CHMP1B helix α1 ablate membrane binding in the presence of $PIP_2$, presumably due to a lack of favorable electrostatic interactions between cationic residues on the protein and anion-rich lipid bilayers (Supplementary Fig. 5).

Finally, due to the pinning of $PIP_2$ by K and R residues and of SDPC by F9 and F13 residues, the diffusion coefficients of lipids in the outer leaflet of the bilayer are dramatically slowed in CG-MD simulations (Extended Data Fig. 9), consistent with recent experimental results that used fluorescence recovery after photobleaching to measure lipid lateral diffusion in CHMP1B/IST1 filaments formed using optical tweezers[30]. This reduction in diffusion coefficients has functional implications for membrane fission as membrane shearing becomes possible as the diffusion rate of lipids decreases[7], and recent work has suggested that CHMP1B/IST1 catalyze membrane fission through a shearing mechanism[30]. The necessity of SDPC and $PIP_2$ for membrane binding and constriction highlights the importance of local lipid composition as a regulator of membrane remodeling.

## Discussion

We combined cryo-EM, heavy-atom-labeled lipid probes and CG-MD simulations to investigate membrane structure and mechanics in ESCRT-III constricted lipid tubules. These efforts revealed how phenylalanine residues from CHMP1B helix 1 displace headgroups to 'score' the surface of the bilayer and create a hydrophobic defect. Flexible, polyunsaturated lipid tails enrich within the furrow created by these Phe residues, where they contort to interact with the hydrophobic side chains. These surface defects coalesce into a helical furrow that may, along with lipid headgroup displacement by F9 and F13, stabilize the bilayer at this high degree of curvature by decreasing lipid density in the convex outer leaflet. We also showed that this high membrane

curvature system generates compositionally and dynamically asymmetric leaflets.

While lipid tail backflipping is possible for any of the lipid tails, polyunsaturated tails are exceptionally flexible[3,45,48] and therefore have low energy barriers to 'backflipping'[45,49,54]. This role for polyunsaturated lipids in stabilizing curvature and filling in packing defects by undergoing curvature-driven shape changes[43,49] is consistent with previous reports that SDPC facilitates membrane fission[3] and that polyunsaturated tails prefer to reside near the lipid–water interface, especially in curved bilayers[3,47,49]. Additionally, polyunsaturated lipids preferentially interact with some membrane proteins via van der Waals interactions[46,55].

Together, our findings on compositional variation within the leaflets of ESCRT-III-formed lipid tubules, and the membrane-remodeling assays with different initial lipid compositions, are consistent with lipid compositions having a prominent effect on the energy required to bend a membrane. Membrane-remodeling machines have adapted accordingly to possess organelle-specific properties, and cells likely govern membrane remodeling by regulating local lipid composition through clustering, transport and lipid metabolism[56].

Structural studies of higher order assemblies of biological macromolecules, which are not defined by the linear sequences of their components, are challenging due to the heterogeneity, disorder and dynamics of these assemblies. Membranes and biomolecular condensates are examples of such systems that are difficult to study with molecular resolution[57–60]. We anticipate that the ability to validate or falsify simulations with heavy-atom cryo-EM will enable new questions to be answered about these critical fluid assemblies, their interactions and the dynamics governing their functions.

## Online content

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

## Methods

### Synthesis and characterization of bromolipids

Bromolipids were synthesized as previously described[61]. Here, 1–100 mg of each lipid was dissolved in $CHCl_3$ to 1–10 mg ml$^{-1}$. The lipid solutions were stirred on ice, and bromine (stoichiometric with the number of double bonds in the lipid) was added dropwise. The solution was stirred on ice in the dark for 1 h. Solvent and excess bromine were removed by application of vacuum overnight in the dark. Brominated lipids were then aliquoted and stored at −80 °C until needed. To maintain accurate lipid stock concentrations, no further purification was performed. Consequently, the brominated lipids also contain some unbrominated lipid, whose quantities were determined by mass spectrometry and nuclear magnetic resonance (NMR) spectroscopy.

Proton NMR ($^1H$ NMR) and carbon NMR ($^{13}C$ NMR) spectra were recorded on a Bruker Avance III HD 400 spectrometer in the UCSF NMR Core Facility using Bruker TopSpin 4. NMR spectra were recorded in $CDCl_3$, except for PIP$_2$-Br that was recorded in 20:10:1 $CDCl_3$:MeOD:$D_2O$. NMR spectra were analyzed with MestReNova v.14.2.1. Electrospray ionization mass spectra were recorded with a Waters SQD2 mass spectrometer in the Stanford University Mass Spectrometry Facility. See Supplementary Fig. 1 for NMR and mass spectra. The average numbers of bromine atoms per lipid molecule, as measured by NMR and mass spectrometry, are shown in Supplementary Table 1. We measured pressure-area isotherms for pure unbrominated and brominated lipids as described in detail previously[62]. Briefly, we used a KSV NIMA KN 2002 (Biolin Scientific) Langmuir–Blodgett system with a 273 cm$^2$ Teflon trough and symmetric Delrin barriers controlled by KSV Nima Attension v.2.3. A piece of Whatman no. 1 filter paper was used as a Wilhelmy plate to monitor the surface pressure in the trough. We added water to the clean trough and used a vacuum line attached to a Teflon tip to remove any contaminants from the liquid surface. Lipids (1 mM in chloroform) were spread onto the surface of the water with a glass microsyringe and solvent was allowed to evaporate for 10 min. The barriers were then compressed at a rate of 10 mm min$^{-1}$ and pressure-area isotherms were collected. All isotherms were recorded at room temperature on a subphase of Milli-Q 18 MΩ water.

### Sample preparation

Membrane-remodeling assays were carried out as previously described to create CHMP1B–IST1 copolymer-bound membrane nanotubes[22]. Small unilamellar vesicles (SUVs) were formed by extrusion. To protect the polyunsaturated lipids from oxidation during the formation of vesicles and subsequent membrane-remodeling reaction, we added 300 nmol of the desired lipid mixture to a glass scintillation vial under a nitrogen atmosphere (see Table 1 and Supplementary Table 2 for lipid mixtures). The solvent was evaporated under a stream of nitrogen while rotating the vial. The resulting film was redissolved in 100 µl chloroform and evaporated again. The vial with the dried lipid film was placed under house vacuum in the dark for 2 h. Next, 250 µl of GF buffer (25 mM Tris pH 8.0 and 125 mM NaCl) was added to the scintillation vial and incubated for 10 min in the dark. The lipid film was next resuspended to form multilamellar vesicles by vortexing. SUVs were formed by extrusion 31 times through a polycarbonate membrane with 50 nm pore size using the Avanti Polar Lipids Mini Extruder. SUVs were either used immediately or aliquoted, snap frozen in liquid nitrogen and stored at −80 °C. To form copolymer-bound nanotubes, SUVs (0.5 mg ml$^{-1}$ in GF buffer) were incubated with 5 mg ml$^{-1}$ CHMP1B (in GF buffer) for approximately 6 h at room temperature in the dark. IST1 (in GF buffer + 5% glycerol) was added to 10 mg ml$^{-1}$ and the solution was incubated overnight at room temperature in the dark.

Samples for negative stain EM and cryo-EM were prepared similarly to previous reports[22]. Briefly, for negative stain EM, 4 µl of the membrane-remodeling mixture were applied to glow-discharged, 200 Cu mesh carbon-coated grids (Electron Microscopy Supplies). Grids were stained with 0.75% (w/v) uranyl formate (Structure Probe, Inc.).

All samples were imaged on an FEI Tecnai T12 120 kV electron microscope equipped with a Gatan UltraScan 895 4k CCD camera. For cryo-EM, 4 µl of membrane-remodeling mixture were pipetted onto glow-discharged R1.2/1.3 Quantifoil 200 Cu mesh grids (Quantifoil) in a Mark IV Vitrobot (FEI). After a 10 s wait time at 19 °C and 100% humidity, grids were blotted for 4 s with Whatman no. 1 filter paper and plunged into liquid ethane. Grids were stored under liquid nitrogen until imaged.

### Cryo-EM imaging

Cryo-EM data were collected on 300 kV FEI Titan Krios or 200 kV FEI Talos Arctica microscopes. The Krios was equipped with a Gatan Bio-Quantum energy filter and Gatan K3 direct electron detector. The Krios was operated at a nominal magnification of ×105,000 with a total dose of 67 e$^-$/A$^2$. The Arctica microscope was operated at a nominal magnification of ×28,000 or ×36,000 with a total dose of 61 e$^-$/A$^2$. The Arctica was equipped with a Gatan K3 direct electron detector. The cameras on both microscopes were operated in correlated double sampling and super resolution modes. Samples with lipid compositions 2–8 and 18 were imaged with the Arctica, while samples with lipid compositions 1 and 10–17 were imaged with the Krios. Images were collected with a 50 µm C2 aperture and 100 µm objective aperture. The defocus was varied between −0.6 and −2 µm. Semiautomated data collection was carried out with SerialEM.

### Data processing

The software programs used to process and visualize data were compiled and configured by SBGrid[63].

We processed the cryo-EM data in a manner similar to that reported previously[22], except that filaments were automatically picked from the micrographs with SPHIRE-crYOLO[64]. The rest of the data processing was performed in RELION v.3.0.8 (ref. [65]). All of the cryo-EM datasets were processed with the pipeline shown in Extended Data Fig. 10 unless otherwise noted. Sample motion was corrected with MotionCor2 with dose weighting, and defocus values were estimated with CTFfind4 (refs. [66],[67]). One round of 2D classification was performed to create a more homogenous data set with similar particle diameters and eliminate particles with poor resolution. A hollow, smooth cylinder was used as an initial model for three-dimensional (3D) auto-refinement without a mask, and 'ignore CTFs until first peak' option for contrast transfer function (CTF) estimation was used. Previously determined helical parameters were used as the initial values in the refinement. Subsequently, these particles underwent a single round of 3D classification to separate 17 and 18 asymmetrical subunits (asu)-per turn structures. Specifically, the number of asymmetrical units, initial twist and initial rise parameters were changed to select for either variant of the filament. During 3D classification, we used a mask that included both the protein and lipid bilayer, whereas further auto-refinements used masks that solely included the protein. High-resolution 3D classes with the desired symmetry were chosen, and each group of classes with the same symmetry went through another round of 3D auto-refinement. Afterward, the refined particles for both the 17 and 18 asu structures went through two rounds of per-particle CTF refinement, with 3D auto-refinement between each round, and the final reconstructions were postprocessed with automatic B factor estimation. The relevant parameters for each step are listed in Supplementary Tables 3–5. We used the CHMP1B–IST1 atomic model previously reported, PDB 6TZ5, which fit all of the 17 subunit per turn reconstructions reported here.

We estimated the local resolution of the lipid bilayer to be roughly 6 Å by gold standard Fourier shell correlation analysis. We created a mask that only encompassed the bilayer and performed an auto-refinement with Fourier shell correlation resolution estimate in RELION. Consequently, we used the same mask to apply a custom 6 Å low-pass filter to the bilayer in the reconstructions while maintaining the high-resolution, roughly 3 Å, features of the protein coat. All the cryo-EM densities shown in the main text have been processed in this

way. We also confirmed that the application of helical symmetry, normalization and low-pass filter did not alter the results of the analyses (Supplementary Fig. 6).

## Analysis of leaflet compositions

We analyzed the leaflet compositions by comparing the increase in intensity of each leaflet for brominated samples versus the unbrominated reference according to the procedure shown schematically in Extended Data Fig. 4a–d. First, we investigated the structure and composition of the membrane–protein contact sites and the region of the bilayer farthest from the protein (gaps between contact sites). To facilitate this comparison, we rotated each $z$ slice of the cryo-EM reconstructions so that membrane–protein contact sites were aligned axially. We then projected the central 30% of each reconstruction along the $z$ axis, improving the signal to noise ratio. We then calculated radial averages and the area of each leaflet curve above 0, which we defined as the solvent background intensity. The areas for each leaflet of the samples without brominated lipids were subtracted from the areas of the respective leaflets of each of the reconstructions with brominated lipids. This yielded the extra scattering in each leaflet due to each brominated lipid. These values were then normalized to the numbers of bromine atoms per molecule (Supplementary Table 1). The values for each leaflet were summed, and the fraction of the total for each brominated lipid was calculated, such that the composition of each leaflet adds to 100 mol%. Second, the bulk, radially averaged compositions of each leaflet were estimated by applying the same procedure but omitting the rotation alignment of the membrane–protein contact sites. Both analyses were performed separately on each of the half maps, and the reported uncertainties are the standard deviations between the analyses from the two half maps. The estimated bulk leaflet compositions are shown in Supplementary Table 6. We also used a commercially available 19-iodo analog of cholesterol to show that the partitioning of this probe between leaflets is similar to that of brominated cholesterol (Supplementary Fig. 7).

## Summary of simulations

Five sets of simulations were conducted. The first set of membrane tubule simulations produced most of the data discussed in the main text (all main text simulation data except Fig. 6). This set consisted of ten replicate simulations using a CG Martini 2.2 model of two turns of the IST1–CHMP1B protein coat (PDB 6TZ5) using a lipid molar composition corresponding to starting lipid composition 1 (Table 1). Full results are presented in Extended Data Fig. 2. The second set was identical to the first, except in silico mutagenesis was used to make the F9A + F13A double mutant of CHMP1B and only three replicates were run (full results in Extended Data Fig. 5). The third set were run using the experimentally derived membrane tubule composition, again with WT CHMP1B (Table 1, composition 17). This more highly anionic lipid mixture was substantially less stable than composition 1, and hence the equilibration procedures and run parameters were adjusted as described in the Supplementary Information (full results in Extended Data Fig. 7). The fourth simulation was run to compare features of the tubules formed in the presence of protein, from simulation set 1, with a tubule of equal curvature but formed without protein. A tubule was formed from spontaneous assembly using a lipid mixture corresponding to composition 1 and run through equilibration and production in the absence of protein. Full results are in Extended Data Fig. 6. The fifth set consisted of one replicate of a simple flat-bilayer simulation corresponding to lipid composition 1. Lipids were allowed to equilibrate between leaflets for several microseconds by opening a pore in the bilayer. See Supplementary Information for a detailed description of this procedure.

## Sets 1 and 2 (CHMP1B/IST1 WT (1) and F9A + F13A CHMP1B mutant (2) with composition 1).

The structure of two turns of the WT CHMP1B–IST1 protein coat (34 subunits of each protein, PDB 6TZ5) was centered in a 30 × 30 × 18 nm simulation box and converted to a CG Martini 2.2 representation. For the F9A + F13A simulations, the protein structure was first edited to contain the mutations and then coarse grained. The protein was fully position restrained throughout simulation to maintain the cryo-EM resolved protein structure in relation to the membrane. Martini model lipids were chosen to correspond to those used in cryo-EM: SDPC (Martini name PUPC), POPS (Martini name POPS) and CHOL (Martini name CHOL) were directly available from the Martini website. A di-oleoyl $PIP_2$ model was constructed from the Martini model 1-palmitoyl-2-oleoyl $PIP_2$ (Martini name POP2) by changing parameters for the second bead of the CG palmitoyl tail from the saturated C type to the unsaturated D type. The total number of lipids was tuned using a set of initial tubule self-assembly simulations as described in Supplementary Information with results in Extended Data Fig. 1, arriving at a total of 1,300 with 754 SDPC, 234 POPS, 234 CHOL and 78 $PIP_2$. Details on lipid placement are described in Supplementary Information. Additionally, the scripts used, as well as a set of example input and output files, for the initial random placement of lipids within the lumen of the protein coat have been deposited on Zenodo (https://doi.org/10.5281/zenodo.7232344). Instructions for their use are in Supplementary Information. The simulation box was then solvated using standard Martini water and ions to neutralize overall system charge and reach 150 mM NaCl. Simulations were run using Gromacs 2018.8, using Martini-recommended run parameters. Reaction-field electrostatics were used with a 1.1 nm Coulomb cutoff. The velocity rescale thermostat was used to maintain a temperature of 320 K, with separate temperature coupling groups for protein, lipids and solvent. The timestep and pressure control method differed between equilibration and production phases and are described in detail in Supplementary Information. One round of minimization was performed before dynamics, which proceeded in three general steps—spontaneous tubule assembly, leaflet equilibration and production—described in Supplementary Information. The results described in the main text are from analysis on the last 2.4 μs of the production phase of each of the ten replicates for the WT simulations. Three replicates were run for each mutant simulation, with analysis also on the last 2.4 μs of the production phase of each replicate. See Supplementary Table 7 for the leaflet compositions of each replicate. Pymol v.2.3.0 was used to visualize simulation frames.

**Set 3 (WT CHMP1B/IST1 with lipid composition 17).** Simulation of WT CHMP1B/IST1 with lipid composition 17 required modifications to the protocol due to instabilities in the membrane that resulted in persistent bubbling across a range of run parameters. These simulations were ultimately run entirely in NVT, as only turning off pressure control reliably avoided bubbling. Full details on the run parameters, other tested combinations and anticipated effects on the results are in Supplementary Information. The local composition results described in main text (Fig. 6) are from analysis on the last 2.4 μs of the production phase of each of the three replicates. See Supplementary Table 7 for the leaflet compositions of each replicate.

**Set 4 (protein-free tubule simulation).** One simulation of a lipid composition 1 tubule of the same curvature without the protein was run. All parameters used and overall setup were identical to simulation set 1, with the exception that isotropic pressure coupling was used, and the leaflet equilibration phase was extended to 30 μs. Production lasted 3 μs, with the last 2.4 μs used for analysis. With semi-isotropic pressure coupling on, and without the protein coat maintaining the tubule shape, the system quickly relaxed to an expanded tubule radius to minimize the curvature energy of the membrane. Therefore, the simulation was run using isotropic pressure coupling, which allowed for fluctuations in the tubule length while maintaining the desired curvature.

**Set 5 (flat bilayer).** We prepared a flat bilayer with 2,023 lipids in the molar ratios of composition 1 in each leaflet in a 36 × 36 × 14 nm box. The simulation was run using a 0.04 ps timestep for 1 µs.

## Simulation analysis

Leaflet assignment was performed with single linkage agglomerative clustering on the Cartesian coordinates of the phospholipid phosphate beads at every frame, unambiguously producing two clusters corresponding to the inner and outer leaflets (Supplementary Fig. 2). For each frame, cholesterols were assigned to a leaflet by determining the nearest phospholipid phosphate to the cholesterol hydroxyl group and assigning the sterol to the same leaflet. See Supplementary Table 8 for leaflet and bilayer thicknesses for each simulated condition.

Cylindrical coordinates $(r, \theta)$ were determined relative to the long axis of the membrane tubule, which was aligned with the $z$ axis of the box. For each frame we determined the optimal position of the tubule center $(x_0, y_0)$ in the $xy$ plane that optimally placed all CHMP1B F9 + F13 (or F9A + F13A) residues along the protein filament equidistant from the cylindrical center line using a least-squares fitting procedure. All atomic coordinates for that snapshot were then calculated in that reference frame. For the protein-free tubule, the mid-line of the cylinder was taken as the $z$ axis of the box.

Headgroup density calculations were performed using the NC3 bead of SDPC, CNO bead of POPS and the C1 bead of $PIP_2$. A 2D Gaussian kernel density estimate on the set of $(\theta, z)$ coordinates from each trajectory was then calculated to compute the heatmaps. The grid spacing for the kernel density estimate was set of 0.835 Å in $z$ and 1° in $\theta$, to correspond with the EM density analysis of the tubule surface. To avoid edge effects, positions within the upper and lower 15% of the borders in $z$ and $\theta$ were mirrored to effectively expand the box, and the kernel density for that expanded box was calculated and then trimmed back down to the unmirrored box dimensions for plotting. Densities for individual trajectories were subsequently averaged and normalized to span the range from 0 to 1.

Radial density profiles for each lipid bead type were calculated for each leaflet within different zones of the membrane surface relative to the protein coat (Supplementary Information and Supplementary Fig. 3) using a Gaussian kernel density estimate on the $r$ coordinates from each trajectory. The bandwidth was set to 0.5 Å, and densities were calculated from 0 to 80 Å with a 0.2 Å grid spacing. Densities were then normalized by $2\pi r \Delta r$ to account for the increasing effective volume at greater $r$, and then multiplied by the number of beads of the specified type within that zone. Thus, densities reflect enrichment or depletion of different lipid species and components. Replicate profiles were averaged to arrive at the final density profile.

Lipid tilt vectors for each tail were calculated by first computing lipid orientation vectors for each tail defined as the vector from the lipid phosphate bead to the last bead of each tail. The bilayer normal was assumed to be the radial component of the cylindrical coordinate system, using $+\hat{r}$ for the inner leaflet and $-\hat{r}$ for the outer leaflet. Hence, tilt values of 0° correspond to lipids with configurations normal to the local membrane plane regardless of leaflet assignment. Lipids were assigned to zones based on the location of the phosphate bead. Per zone angle distributions for each lipid type and tail were calculated across each replicate using a Gaussian kernel density estimate spanning 0 to 180° with a bandwidth of 0.05° and a grid spacing of 0.9°. Replicate profiles were averaged to arrive at the final density profile.

Local compositions per zone were calculated by gathering all nonheadgroup beads within each zone and counting the number of beads present from each lipid species. The total number of beads per species was then normalized by the number of beads that make up a whole lipid of each individual lipid type in the model, excluding the headgroups, to get a fractional molar number of each type of lipid present (that is, 12 beads for SDPC, 11 for POPS, eight for CHOL and 11 for $PIP_2$). Headgroups were excluded since they have an outsized

effect for $PIP_2$ lipids (five beads per headgroup versus one for SDPC and POPS). The normalized molar values for each lipid type were then summed and used to calculate local molar compositions per zone and leaflet. Compositions from independent trajectories were averaged to arrive at final reported values.

## Reporting summary

Further information on research design is available in the Nature Portfolio Reporting Summary linked to this article.

## Data availability

The cryo-EM maps have been deposited into the Electron Microscopy Data Bank (accession numbers EMD-27991, EMD-28694–28719 and EMD-28722). The motion-corrected cryo-EM micrographs have been deposited in the EMPIAR database (https://www.ebi.ac.uk/empiar/) with accession number EMPIAR-11277. PDB 6TZ5 was used for simulations and data analysis. Source data are provided with this paper.

## Code availability

Code for setting up and analyzing molecular dynamics simulations has been deposited on Zenodo (https://doi.org/10.5281/zenodo.7232344).

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

## Acknowledgements

We thank members of the Frost and Grabe laboratories for helpful discussion, especially P. Thomas for computational assistance, H. Nguyen and H. Aydin for assistance with ESCRT-III proteins, and D. Argudo and N. Bethel for early discussions concerning membrane tubule mechanics and simulations. We also thank the UCSF Center for Advanced Cryo-EM, including D. Bulkley, G. Gilbert, E. Tse, M. Harrington and Z. Yu. This work was supported by a National Institutes of Health (NIH) postdoctoral fellowship to J.L. (grant no. 4T32HL007731-28), grant nos. NIH R01 GM117593 and NIH R01 GM137109 (M.G.), NIH P50 GM082545 (A.F.) and NIH 1DP2-GM110772 (A.F.), and hardware for simulations was provided by grant no. NIH R01GM089740 (M.G.). Simulations were also carried out on the UCSF Wynton Cluster made possible through grant nos. NIH 1S10OD021596 and 1S10OD020054-011. Microscopes are supported by grant nos. NIH 1S10OD021741-01

and NIH S10OD026881-01. A.F. is further supported by a Faculty Scholar grant from the HHMI and is a Chan Zuckerberg Biohub investigator. F.R.M. is supported by a postdoctoral fellowship from The Jane Coffin Childs Memorial Fund for Medical Research. Structural biology applications used in this project were compiled and configured by SBGrid. Part of this research was supported by the UCSF Sandler Program for Breakthrough Biomedical Research. The graphical processing units used for cryo-EM data processing were donated by the NVIDIA Corporation. Part of this work was performed at the Stanford Nano Shared Facilities, supported by the National Science Foundation (grant no. ECCS-1542152). Mass spectrometry was performed at the Vincent Coates Foundation Mass Spectrometry Laboratory, Stanford University Mass Spectrometry, supported in part by grant no. NIH P30 CA124435 using the Stanford Cancer Institute Proteomics/Mass Spectrometry Shared Resource.

## Author contributions

F.R.M. synthesized and characterized bromolipids, expressed and purified proteins, performed biophysical and cryo-EM experiments, and analyzed cryo-EM data. A.M. analyzed cryo-EM data. J.L., M.T. and A.M. performed and analyzed molecular dynamics simulations. M.G. and A.F. supervised the research and obtained funding. All authors discussed results and prepared the manuscript.

## Competing interests

A.F. is a shareholder and an employee of Altos Labs, Inc. and a shareholder and consultant for Relay Therapeutics, LLC. F.R.M. and A.M are employees of Altos Labs, Inc.

## Additional information

**Extended data** is available for this paper at https://doi.org/10.1038/s41594-022-00898-1.

**Correspondence and requests for materials** should be addressed to Michael Grabe or Adam Frost.

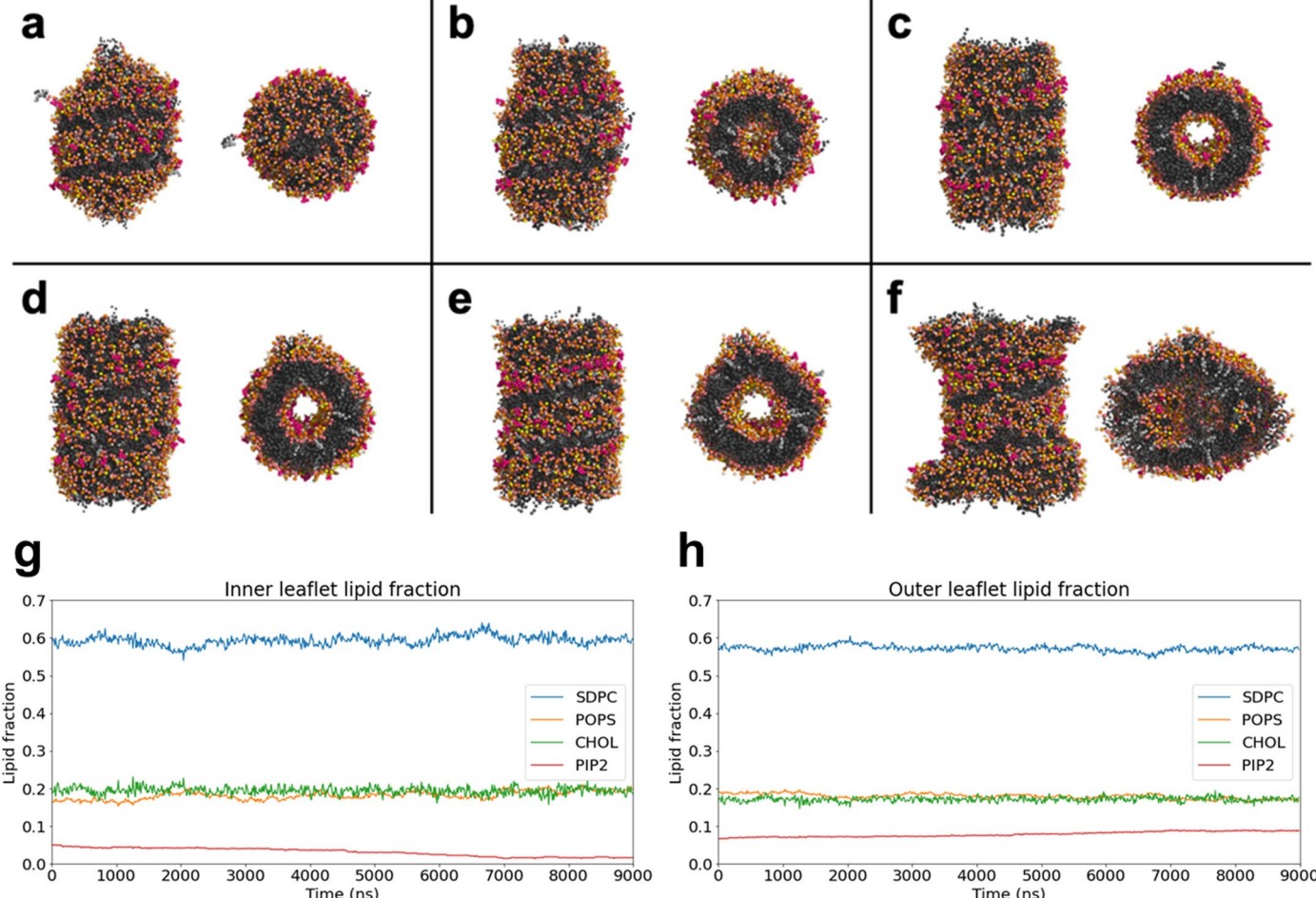

**Extended Data Fig. 1 | Optimization of lipid density and leaflet equilibration in CG-MD simulations.** (a–f) Final frame of tubule from side (left) and top-down (right) from simulations tuning total lipid count. Solvent and protein removed for clarity. Lipid composition was composition 1, with total lipid amounts (**a**) 1017, (**b**) 1187, (**c**) 1271, (**d**) 1314, (**e**) 1356, and (**f**) 1695. Cholesterol in light gray, lipid tails in dark gray, glycerol in light pink, phosphates in orange, SDPC headgroups in beige, POPS headgroups in yellow, and $PIP_2$ headgroups in bright pink. Insufficient lipids result in vesicle formation (**a**) or tubule with varying degrees of neck narrowing away from the protein (**b**, **c**); whereas an over-packed tubule bulges outward away from the protein (**e**) or fills the tubule lumen with lipids (**f**). A lipid count of 1300 (**d**) was used for production simulations since this resulted in a consistent tubule radius along the z axis with a clearly defined and solvated tubule lumen. These simulations employed a protein coat (PDB ID 3JC1) with opposite handedness to the coat used for all simulations reported in the main text (PDB ID 6TZ5), but nearly identical dimensions. These simulations used an isotropic Berendsen barostat for pressure coupling and were run with a 0.03 ps timestep for 6 μs, while all other parameters were identical to those described elsewhere in Methods). (**g**, **h**) Leaflet compositions over time from one replicate of WT CHMP1B-IST1 composition 1 simulation during leaflet equilibration (procedure described in SI Section 13). Leaflet assignment was determined using a simple $r$ cutoff (r < 42 Å inner, else outer) applied to the lipid center of mass, since the clustering algorithm performed poorly in the presence of the equilibration pore due to the connection between the two leaflets. $PIP_2$ was found to typically require the longest time to equilibrate, likely due to its low total number in the lipid mix and strong electrostatic attraction to the protein coat. Equilibration was run for 9 μs for composition 1 simulations and 6 μs for composition 17 simulations, as composition 17 replicates more quickly converged to similar leaflet compositions. Data for graphs in g, h are available as source data.

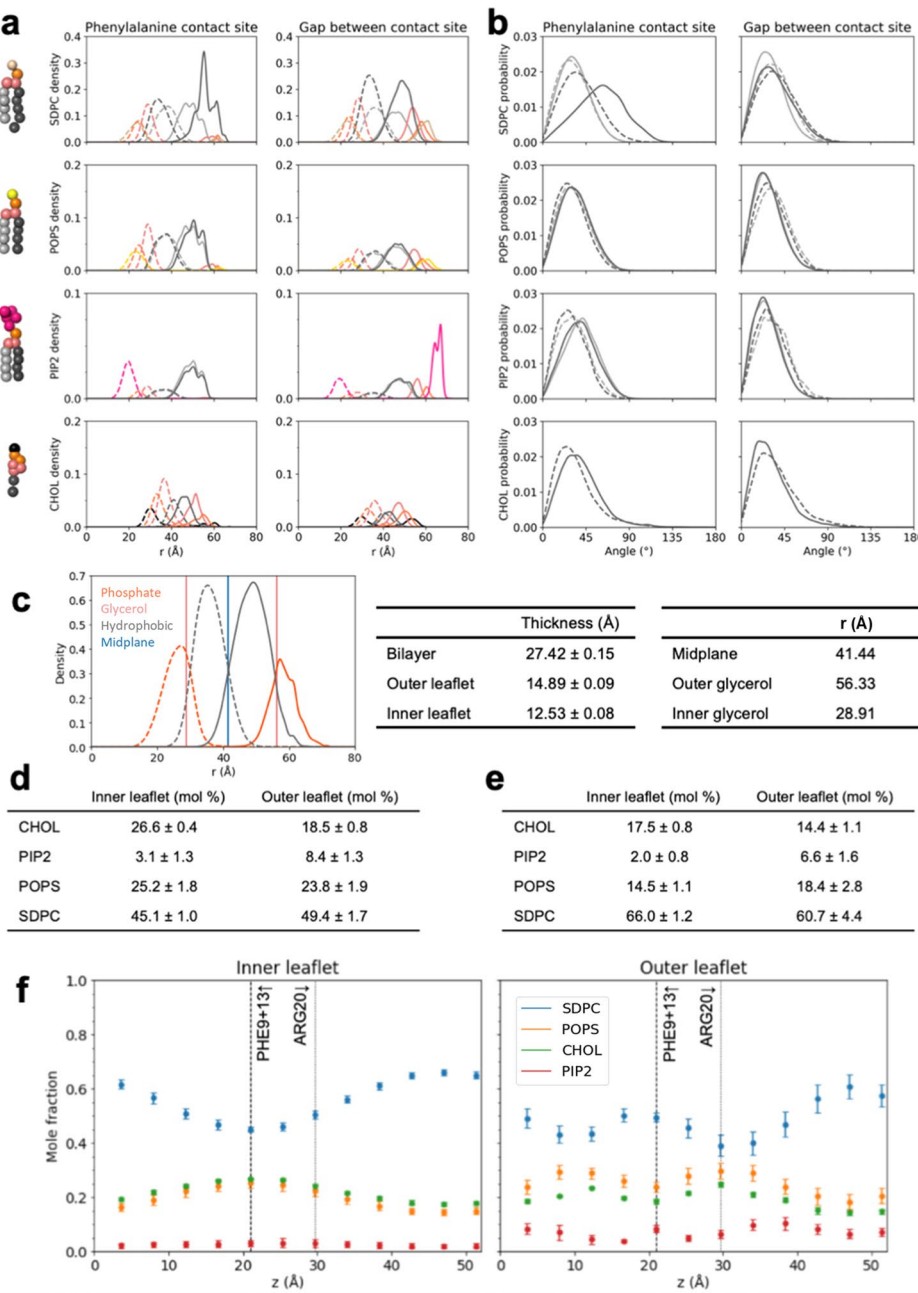

**Extended Data Fig. 2 | Lipid bilayer structure from CG-MD simulations with wild type CHMP1B-IST1 and lipid composition 1.** (a, b) Full radial density (a) and tail angle data (b) at the phenylalanine contact site (left panels) and the gap between the contact site (right panels). Radial density profiles are color coded by lipid segment. Phospholipids: sn-1 tail (light grey), sn-2 tail (dark grey), glycerol (light pink), phosphate (orange), and headgroups (SDPC-beige, POPS-yellow, PIP₂-hot pink). Cholesterol: hydroxyl (black), R1 and R2 ring beads (orange), R3-R5 ring beads (light pink), and aliphatic tail (dark grey). Line style denotes outer (solid) and inner (dashed) leaflet densities. Note y-axis scale varies by lipid type due to abundance. Tail angle distribution plots (b) use the same color scheme and line styles as (a), and all distributions are normalized to 1. (c) Bilayer and leaflet thicknesses determined from mean radial lipid positions. Radial densities of hydrophobic (gray curves) and hydrophilic (orange curves) membrane components for inner leaflet (dashed lines) and outer leaflet (solid

lines). Vertical lines denote the position of the midplane (blue) and mean glycerol inner and outer leaflet planes (pink) for calculating leaflet thicknesses. Bilayer and leaflet thicknesses are summarized in the center table, and mean locations of midplane and glycerol planes are in the right table. (d, e) Mean local compositions at the phenylalanine contact site (d) and at the gap between the contact site (e). The ± value is the standard deviation in mole fraction across the 10 independent replicates. (f) Local compositions per leaflet per zone, plotted as the axial z position of the zone to correspond with Fig. 3f (main text). Each lipid is color coded: SDPC (blue), POPS (orange), CHOL (green), and PIP₂ (red). Vertical dashed and dotted lines denote locations of membrane facing CHMP1B residues for reference. See Fig. S3 for zone definitions. Error bars represent ± 1 standard deviation between replicates. Strong variations in local composition are seen in each leaflet primarily for SDPC, the majority component. Data for graphs in a-c, f are available as source data.

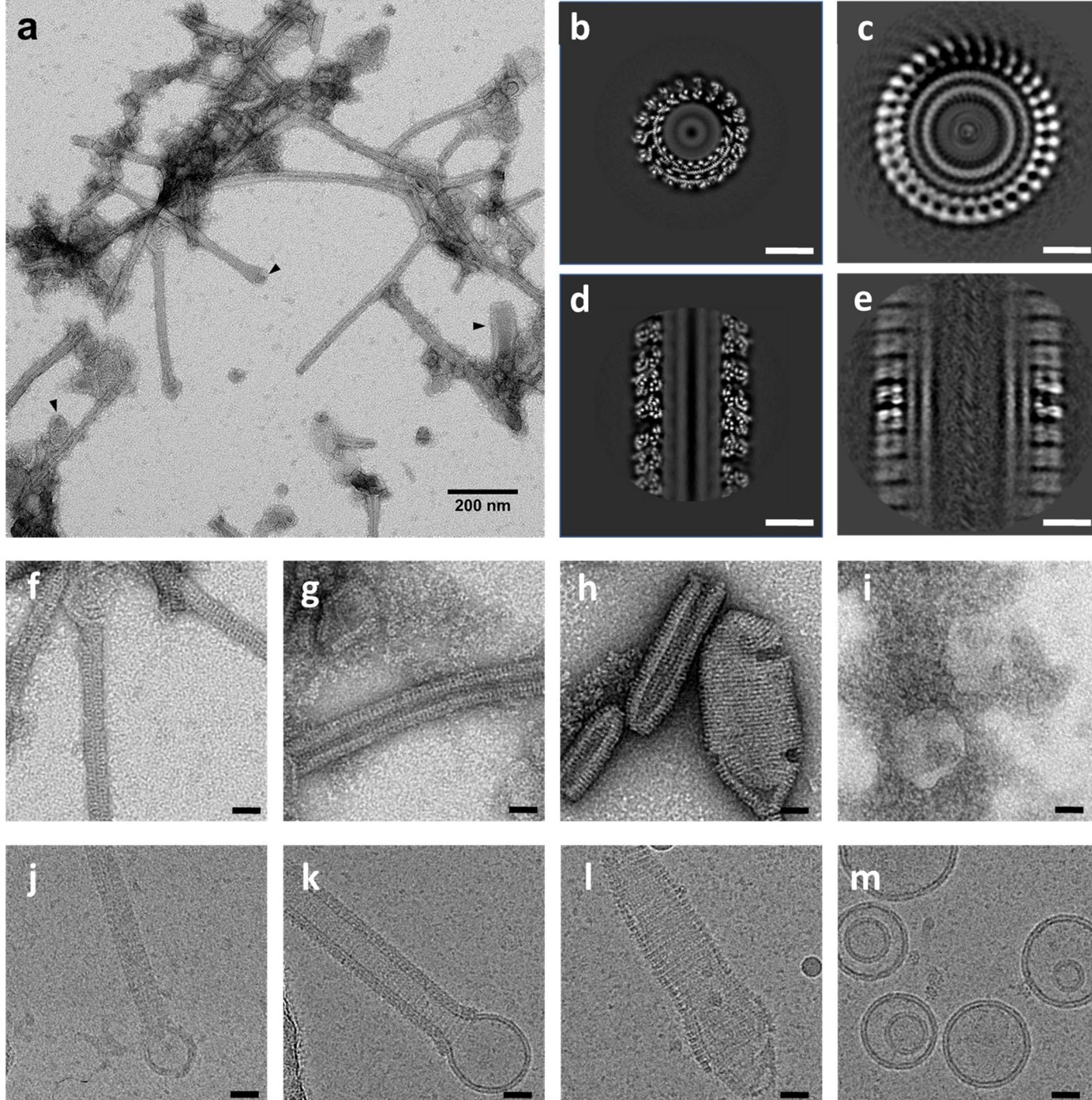

**Extended Data Fig. 3 | Negative stain TEM and cryo-EM of CHMP1B/IST1 filaments with different lipid compositions.** (**a**) Negative stain TEM micrograph of CHMP1B/IST1 filaments. Black arrows identify examples of vesicular remnants protruding from the ends of filaments, vesicles with no protein bound, or unusually wide filaments. (**b**) Horizontal slice of a cryo-EM reconstruction of the 17 subunit per turn structure with lipid mixture 1 (**c**) Vertical slice of a cryo-EM reconstruction of the 17 subunit per turn structure with lipid mixture 1. (**d**) Horizontal slice of a cryo-EM reconstruction of the 34 subunit per turn structure with lipid composition 3. (**e**) Vertical slice of a cryo-EM reconstruction of the 34 subunit per turn structure with lipid composition 3. Negative stain micrographs of (**f**) Lipid composition 1. g) Lipid Mixture 3. (**h**) Lipid composition 9. (**i**) lipid composition 2. Cryo-EM micrographs of (**j**) Lipid composition 1. (**k**) Lipid composition 3. (**l**) Lipid composition 9. (**m**) Lipid composition 2. Scale bars represent 200 nm (**a**), 10 nm (**b**–**e**), and 25 nm (**f**–**m**). Negative stain micrographs are representative of approximately 10 recorded for each sample. Cryo-EM micrographs are representative of approximately 500 collected for each sample.

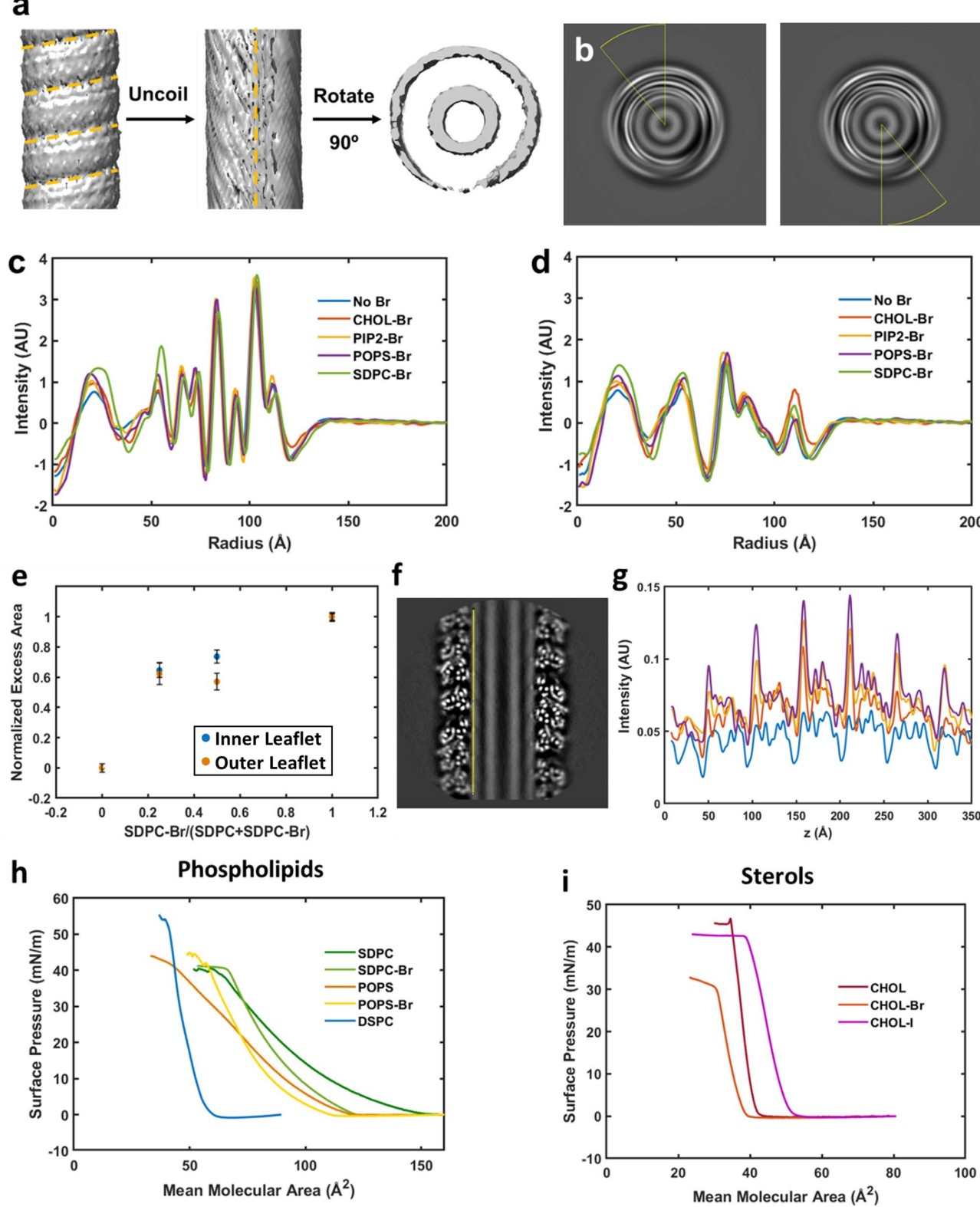

**Extended Data Fig. 4 | See next page for caption.**

**Extended Data Fig. 4 | Compositional analysis of lipid bilayers in cryo-EM reconstructions and biophysical characterization of brominated lipids. (a)** Analysis of the structure and composition of the membrane-protein contact sites and the gaps between contact sites. To improve the signal to noise ratio, each Z slice of the cryo-EM reconstruction was rotated to align it with the previous frame, resulting in an 'uncoiled' nanotube that was projected in Z. **(b)** Radial averages for 30 ° at the contact sites (left) and between the contact sites (right), where the protein is farthest from the bilayer. Radial averages for cryo-EM reconstructions at the contact sites **(c)** and at the gaps between contact sites **(d)**. Cryo-EM reconstructions from samples in which different fractions of the SDPC was brominated. **(e)** Quantification of the excess intensity in each leaflet, with normalization per leaflet. Reducing the SDPC-Br concentration results in attenuated Br scattering in each leaflet. The reduction is not linear with SDPC-Br concentration, suggesting that SDPC-Br is somewhat enriched in the lipid bilayer nanotubes relative to SDPC. Data are plotted as the mean from independent calculations from each of the two half maps, and error bars represent the standard deviation from the independent half maps. **(f)** Visualization of the SDPC-Br titration along a Z line scan of the outer leaflet from vertical slices of the cryo-EM reconstructions **(g)**. **(h, i)** Langmuir pressure-area isotherms show that brominated lipids behave similarly to their unbrominated analogs in lipid monolayers. Although brominated lipids are less compressible than the unsaturated lipids, they are much more compressible than saturated lipids like DSPC. Data for graphs in **c**–**e**, **g**–**i** are available as source data.

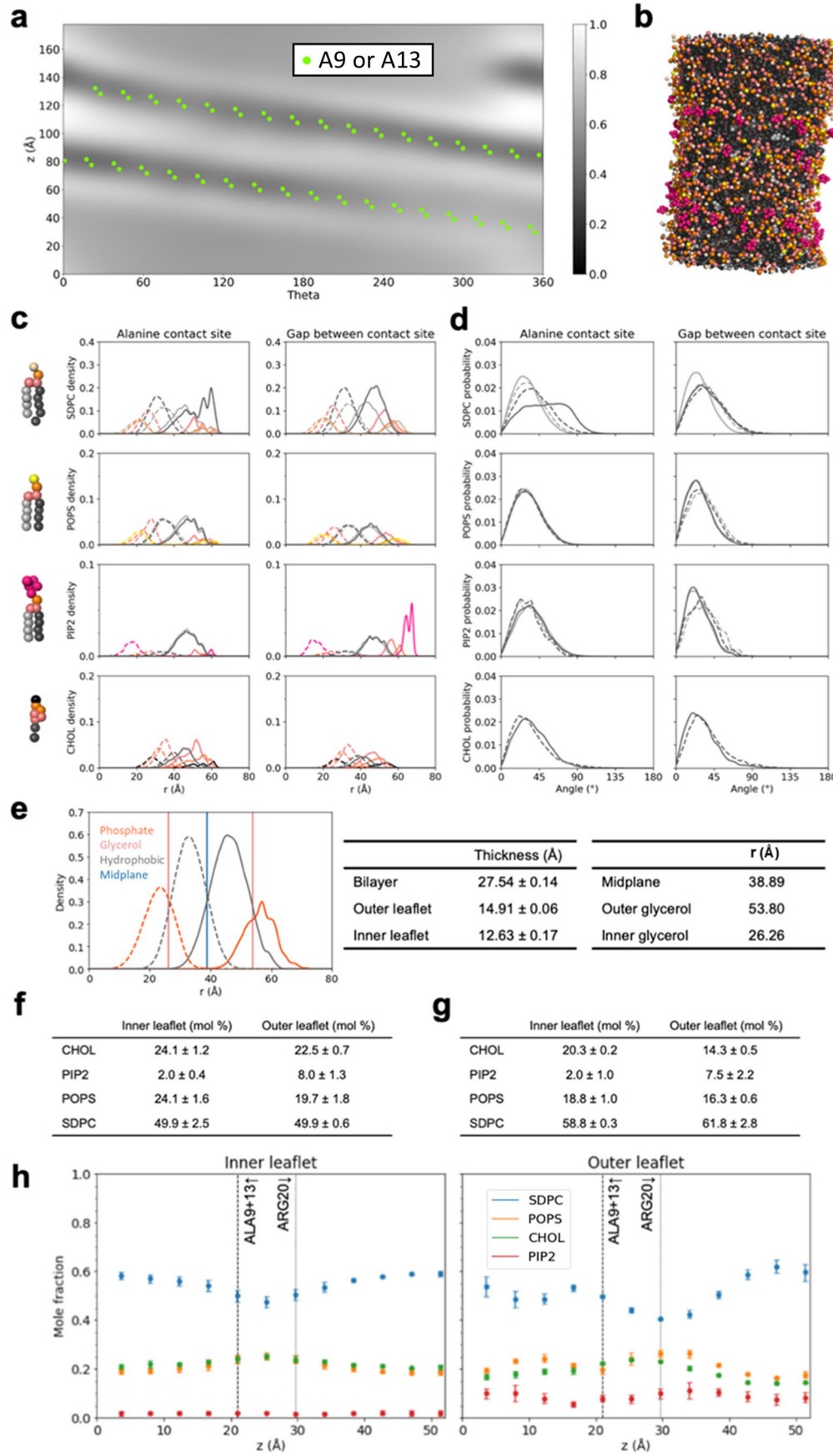

**Extended Data Fig. 5 | See next page for caption.**

**Extended Data Fig. 5 | Lipid bilayer structure from CG-MD simulations with F9A + F13A CHMP1B and lipid composition 1.** (**a**) Average outer leaflet headgroup density represented as a two-dimensional ($\theta$, $z$) density from three independent replicates. F9A + F13A positions denoted by green dots. (**b**) Snapshot of the outer tubule surface, with protein and solvent hidden (color coding same as Extended Data Fig. 1a–f). The smaller alanine side chains appear less able to maintain a persistent hydrophobic defect than the WT (main text Fig. 1h). (**c**, **d**) Full radial density (**c**) and tail angle data (**d**) at the mutated alanine contact site (left panels) and the gap between the contact site (right panels). Formatting corresponds to Extended Data Fig. 2a, b. At the mutated alanine contact site in the outer leaflet, phospholipid headgroups are pushed inward radially, consistent with a putative elastic deformation of the bilayer, while the polyunsaturated tail of SDPC displays reduced backflipping compared to WT (Extended Data Fig. 2a). For tail angle distributions (**d**), outer leaflet lipid tails are more tilted at the alanine contact site than elsewhere, though less than at WT

F9 + F13 tubule site (Extended Data Fig. 2b). (**e**) Bilayer and leaflet thicknesses determined from mean radial lipid positions, formatting corresponds to Extended Data Fig. 2c. Bilayer and leaflet thicknesses in the center table; mean locations of midplane and glycerol planes in the right table. Leaflet thickness are nearly identical to the WT composition 1 simulation values (Extended Data Fig. 2c), though each plane is shifted to slightly lower values of *r*. (**f**, **g**) Mean local compositions at the alanine contact site (**f**) and at the gap between the contact site (**g**). The ± value is the standard deviation across the 3 independent replicates. (**h**) Local compositions per leaflet per zone, plotted as the axial z position of the zone to correspond with Fig. 3f (main text). Color coding and formatting identical to Extended Data Fig. 2f. Error bars represent ±1 standard deviation between replicates. The variations in local composition appear to follow similar trends as in the WT composition 1 simulations (Extended Data Fig. 2f), though at smaller magnitude. Data for graphs in **a**, **c**–**e**, h are available as source data.

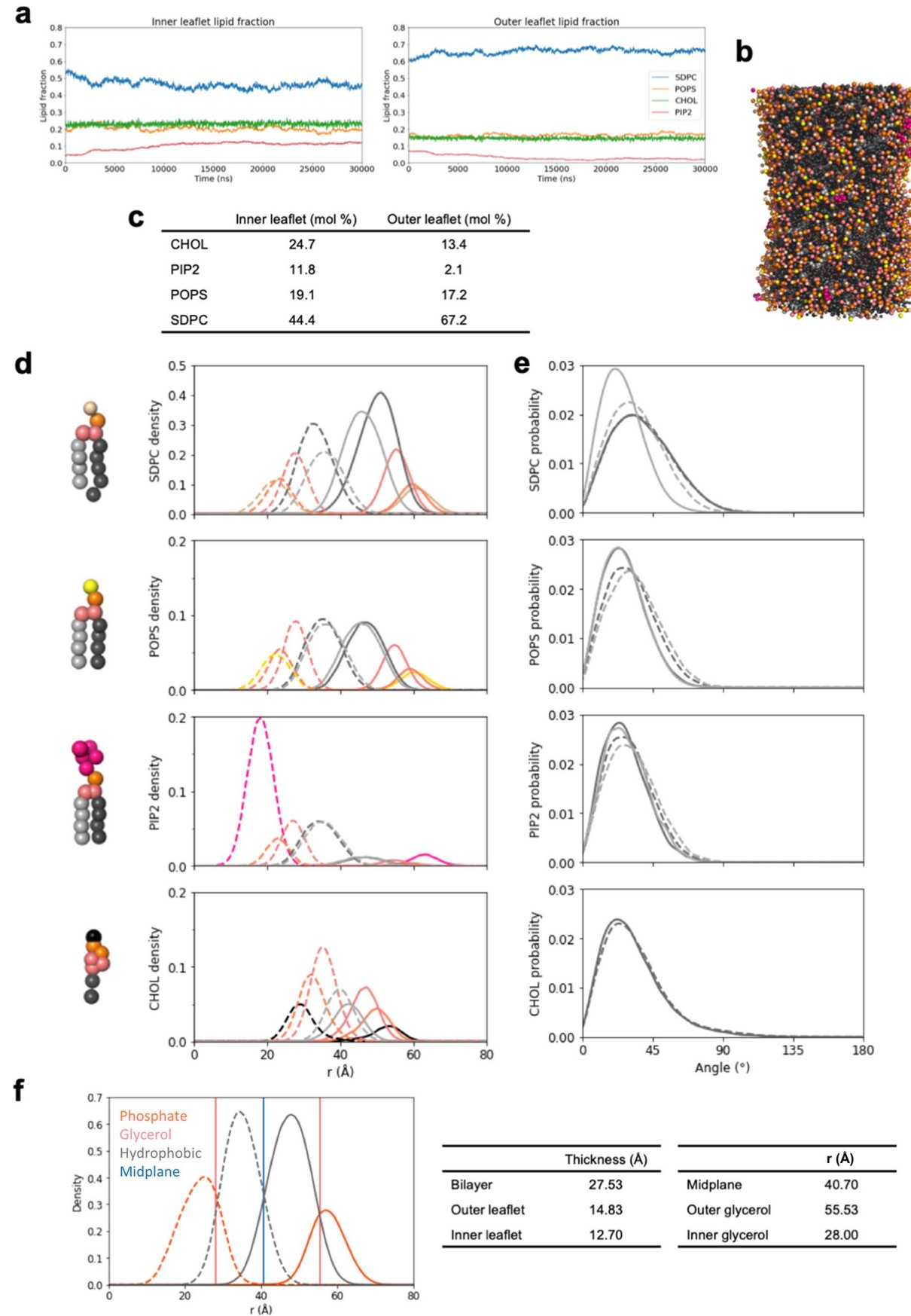

**Extended Data Fig. 6 | See next page for caption.**

**Extended Data Fig. 6 | Protein-free lipid bilayer structure from CG-MD simulations of lipid composition 1.** (**a**) Compositions of the inner (left) and outer (right) leaflets over time during open-pore leaflet equilibration phase. (**b**) Snapshot of the outer tubule surface from the last frame from the production stage with solvent hidden (see Extended Data Fig. 1 for color coding). (**c**) Overall leaflet compositions calculated from production data. Relative to the WT CHMP1B-IST1 lipid composition 1 simulations (Table S7), there is stronger SDPC partitioning to the outer leaflet, stronger CHOL partitioning to the inner leaflet, and an inversion of the $PIP_2$ partitioning now to the inner rather than the outer leaflet. d-e) Full radial density (**d**) and tail angle data (**e**). Radial density profiles correspond to different segments of the lipid, color coded as shown on the CG representations at the left. See Extended Data Fig. 2 for color coding.

Outer leaflet densities are drawn as solid lines; inner leaflet densities are drawn as dashed lines. Note y-axis scale varies by lipid type due to abundance. In the absence of the protein, lipids are relatively normally oriented and there is no extreme backflipping even by the polyunsaturated tail of SDPC. That tail remains slightly shifted away from the bilayer center due to its flexibility but does not show substantial density past the phosphate peak (compare to Extended Data Fig. 2a). Tail angle distributions (**e**) reflect this as well. Color scheme and line styles follow the same scheme as (**d**), and all distributions are normalized to 1. (**f**) Characterization of bilayer and leaflet thicknesses. Bilayer and leaflet thicknesses are shown in the center table, and mean locations of the midplane and glycerol planes are at the right. Data for graphs in **a**, **d**–**f** are available as source data.

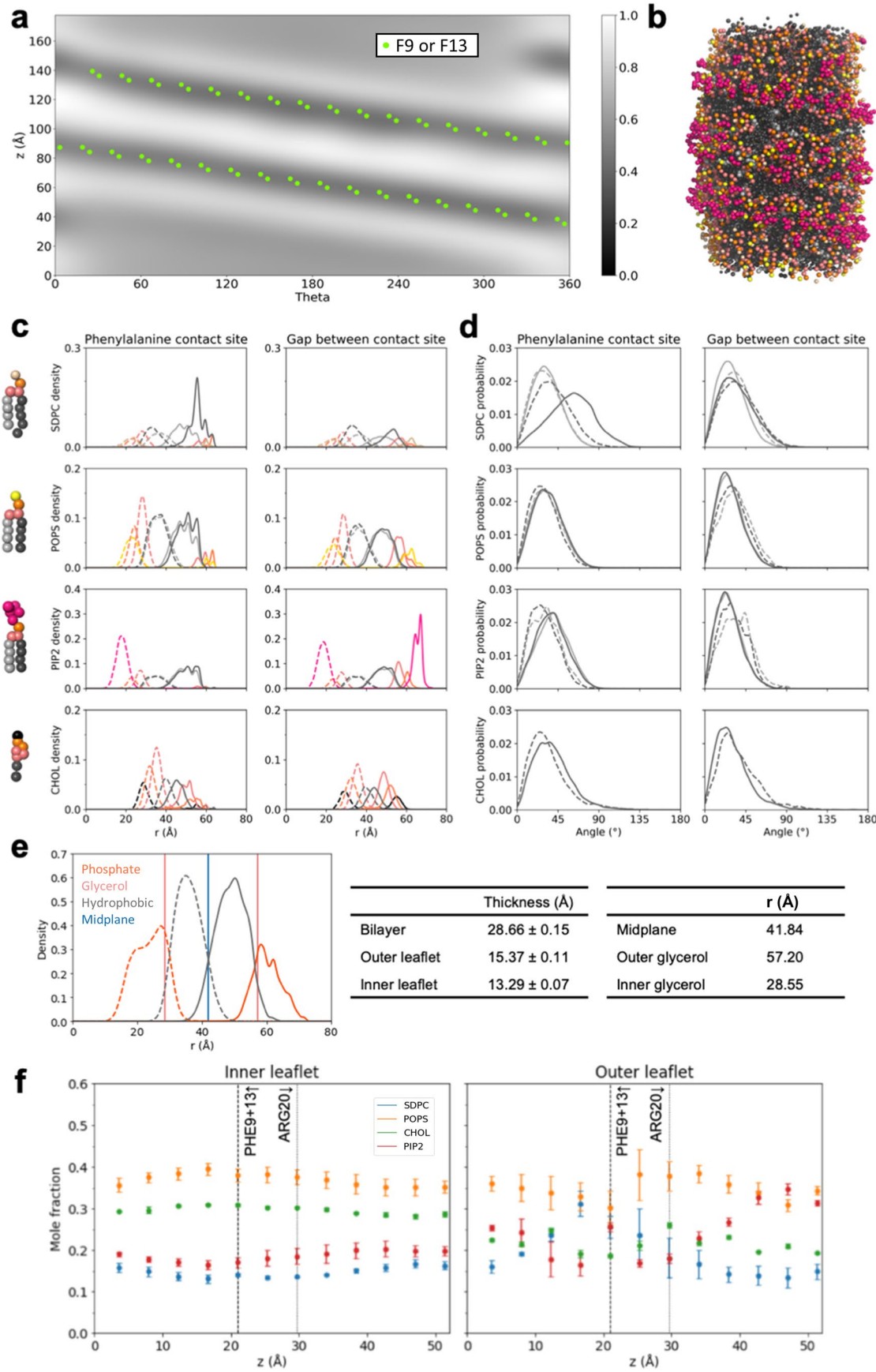

**Extended Data Fig. 7 | See next page for caption.**

**Extended Data Fig. 7 | Lipid bilayer structure from CG-MD simulations with wild type CHMP1B-IST1 and lipid composition 17 simulations.** (**a**) Average outer leaflet headgroup patterning represented as a two-dimensional (θ, z) density from three independent replicates. (**b**) Representative snapshot of the outer tubule surface, with protein and solvent hidden (see Extended Data Fig. 2 for color coding). The scale and persistence of the hydrophobic defect appear similar to the WT simulations using Composition 1 (Fig. 1h). (**c**, **d**) Full radial density (**c**) and tail angle data (**d**) at the phenylalanine contact site (left panels) and the gap between the contact site (right panels). Radial density profiles correspond to different segments of the lipid, color coded as shown on the CG representations at the left. See Extended Data Fig. 2 for color coding. Outer leaflet densities are drawn as solid lines; inner leaflet densities are drawn as dashed lines. Note y-axis scale varies by lipid type due to abundance. Main structural features seen in the WT protein + lipid composition 1 simulations

(Extended Data Fig. 2a, b) are preserved despite the varied composition. Tail angle distributions (**d**) behave similarly as well. Color scheme and line styles follow the same scheme as (**c**), and all distributions are normalized to 1. (**e**) Bilayer and leaflet thicknesses determined from mean radial lipid positions. Bilayer and leaflet thicknesses are summarized in the center table, and mean locations of midplane and glycerol planes are in the right table. The leaflets similarly show differential thinning, and the bilayer is only marginally thicker than that of the WT composition 1 simulations (Extended Data Fig. 2c), perhaps indicating that the reduction in overall SDPC content for composition 17 produces a less flexible, thicker membrane. (**f**) Local compositions per leaflet per zone, plotted as the axial z position of the zone to correspond with Fig. 3f (main text). See Fig. S3 for zone definitions. Error bars represent ±1 standard deviation between replicates. Data for graphs in a, **c**–**e**, **f** are available as source data.

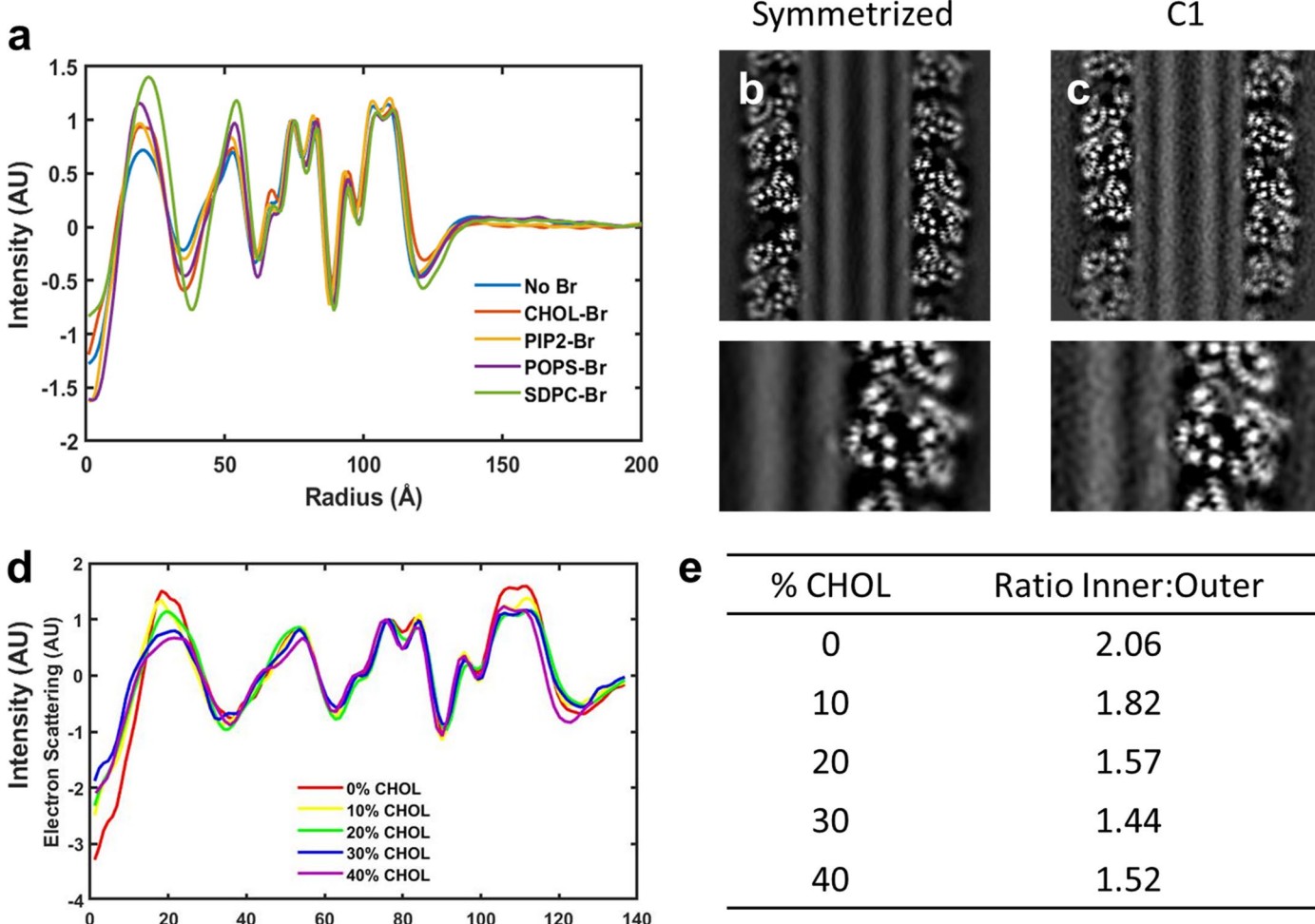

**Extended Data Fig. 8 | Comparison of cryo-EM reconstructions of CHMP1B/IST1 filaments with different lipid compositions.** (**a**) Normalized radial profiles of cryo-EM reconstructions with and without brominated lipids. The central 30 % of each cryo-EM reconstruction was projected along the helical axis, and the radial average was computed. (**b**) Vertical slices through the cryo-EM reconstruction with SDPC-Br with helical symmetry applied and (**c**) without helical symmetry applied. (**d**) Radial profiles of cryo-EM reconstructions with different cholesterol concentrations show overall enrichment of cholesterol in the inner leaflet. (**e**) Ratio of areas under each leaflet peak from (**d**) show a decrease in the intensity of the inner leaflet peak as the cholesterol concentration increases, consistent with cholesterol's lack of electron dense phosphates. Data for graphs in a, d are available as source data.

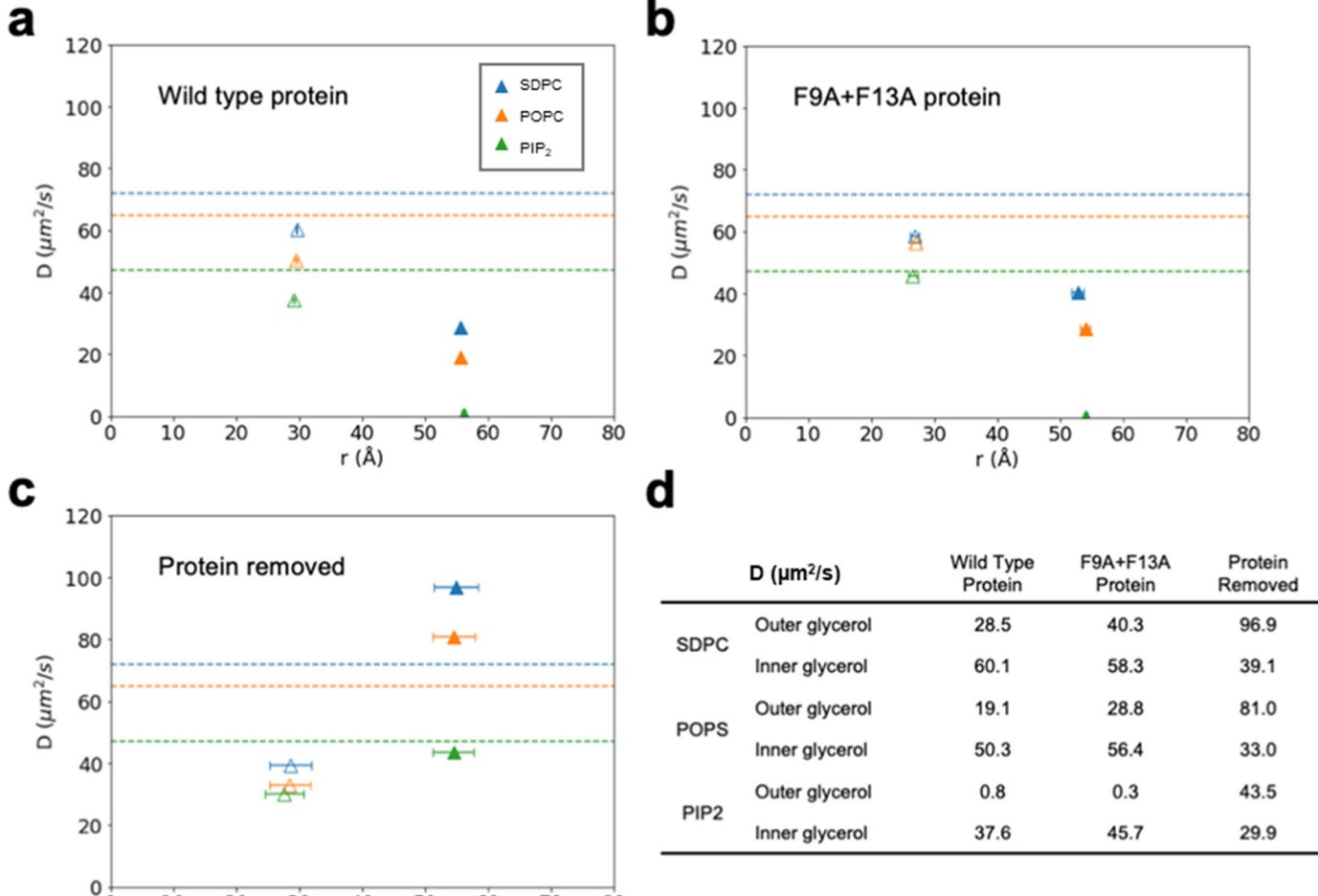

| D (µm²/s) | | Wild Type Protein | F9A+F13A Protein | Protein Removed |
|---|---|---|---|---|
| SDPC | Outer glycerol | 28.5 | 40.3 | 96.9 |
| | Inner glycerol | 60.1 | 58.3 | 39.1 |
| POPS | Outer glycerol | 19.1 | 28.8 | 81.0 |
| | Inner glycerol | 50.3 | 56.4 | 33.0 |
| PIP2 | Outer glycerol | 0.8 | 0.3 | 43.5 |
| | Inner glycerol | 37.6 | 45.7 | 29.9 |

**Extended Data Fig. 9 | Lipid diffusion rates from CG-MD simulations of WT and F9A + F13A CHMP1B with lipid composition 1.** (a–c) Phospholipid diffusion rates in the WT (**a**), F9A + F13A (**b**), and protein free (**c**) simulations plotted against the mean radial coordinate of the lipid's second glycerol bead (GL2). Diffusion rates calculated in a flat bilayer are represented as horizontal dashed lines. Inner leaflet values are shown with hollow markers and outer leaflet markers are filled. Horizontal error bars at left are ±1 standard deviation of the mean time-averaged radial coordinate across ten trajectories, and at right are ±1 standard deviation of radial coordinates across time in the protein-free trajectory. (**d**) Diffusion rates in the presence of protein are similar between the WT and mutant CHMP1B simulations, with outer leaflet lipids—especially PIP₂— showing dramatically slowed diffusion rates relative to inner leaflet lipids. This reduction in diffusion is likely due to electrostatic interactions with positively charged CHMP1B residues, which transiently pin charged lipids like PIP₂, and

the headgroup exclusion region at the membrane-protein contact site (Fig. 1h), which largely prevents lipids from crossing the helical furrow formed by these residues. The slight increase in diffusion for outer leaflet SDPC and POPS in the F9A + F13A simulations may reflect the lessened headgroup exclusion (Extended Data Fig. 5a). With the protein removed, each lipid shows faster diffusion in the outer leaflet, as expected from the tubule geometry[68]. We expect that the reduction in inner leaflet diffusion rates for the no-protein case relative to the with-protein simulations is attributable to higher amounts of saturated tails in the inner leaflet (Extended Data Fig. 6c vs. Table S7). Faster outer leaflet diffusion in the absence of protein suggests a friction-like effect from the protein. Finally, increased diffusion rates in the outer leaflet of the protein-free tubule relative to the flat bilayer suggests that the increase in area-per-lipid increases diffusion rates. Data for graphs in a-c are available as source data.

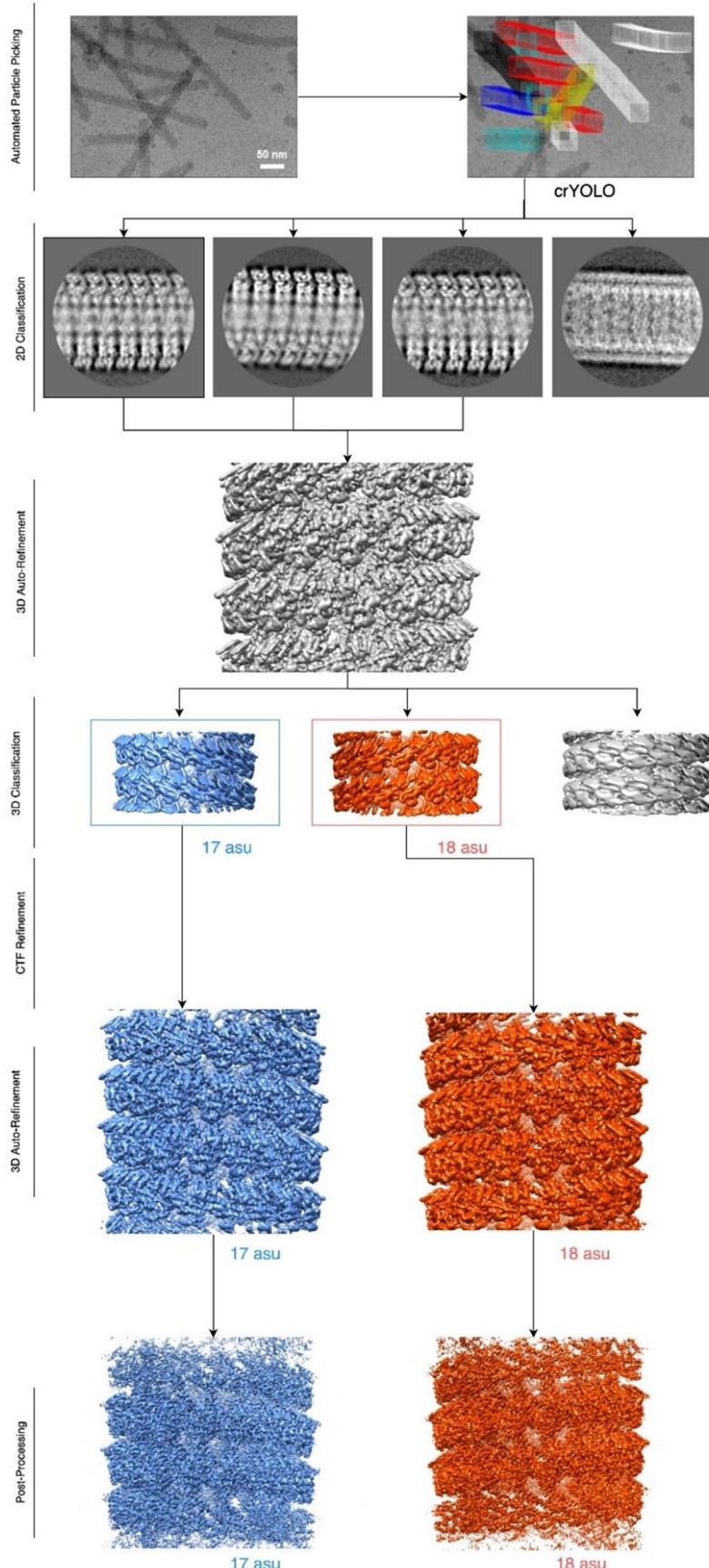

**Extended Data Fig. 10 | Cryo-EM data processing pipeline.** All steps were performed in RELION 3.08 except particle picking, which was performed with SPHIRE-crYOLO. See Tables S3–5 for data processing parameters and the number of particles selected at each step, as well as microscope parameters.

# Reporting Summary

## Statistics

For all statistical analyses, confirm that the following items are present in the figure legend, table legend, main text, or Methods section.

| n/a | Confirmed | |
|---|---|---|
| ☐ | ☒ | The exact sample size (*n*) for each experimental group/condition, given as a discrete number and unit of measurement |
| ☐ | ☒ | A statement on whether measurements were taken from distinct samples or whether the same sample was measured repeatedly |
| ☒ | ☐ | The statistical test(s) used AND whether they are one- or two-sided *Only common tests should be described solely by name; describe more complex techniques in the Methods section.* |
| ☒ | ☐ | A description of all covariates tested |
| ☐ | ☒ | A description of any assumptions or corrections, such as tests of normality and adjustment for multiple comparisons |
| ☐ | ☒ | A full description of the statistical parameters including central tendency (e.g. means) or other basic estimates (e.g. regression coefficient) AND variation (e.g. standard deviation) or associated estimates of uncertainty (e.g. confidence intervals) |
| ☒ | ☐ | For null hypothesis testing, the test statistic (e.g. *F*, *t*, *r*) with confidence intervals, effect sizes, degrees of freedom and *P* value noted *Give P values as exact values whenever suitable.* |
| ☒ | ☐ | For Bayesian analysis, information on the choice of priors and Markov chain Monte Carlo settings |
| ☒ | ☐ | For hierarchical and complex designs, identification of the appropriate level for tests and full reporting of outcomes |
| ☒ | ☐ | Estimates of effect sizes (e.g. Cohen's *d*, Pearson's *r*), indicating how they were calculated |

*Our web collection on statistics for biologists contains articles on many of the points above.*

## Software and code

Policy information about availability of computer code

| Data collection | Electron microscopy data was collected using SerialEM 3.8. NMR data was collected with Bruker TopSpin 4. Langmuir trough isotherms were collected with KSV Nima Attension 2.3. Simulations were performed with Gromacs 2018.8 and Pymol 2.3.0. |
|---|---|
| Data analysis | Data was analyzed with MotionCor2, RELION 3.0.8, CTFFind4, ImageJ 1.53c, MestReNova 14.2.1, UCSF Chimera 1.15, Matlab R2019, Python 3.7.7, MDAnalysis 1.0.0, and SPHIRE-crYOLO 1.7.6. The software is referenced in the Methods section. Custom code is deposited with Zenodo (10.5281/zenodo.7232344). |

For manuscripts utilizing custom algorithms or software that are central to the research but not yet described in published literature, software must be made available to editors and reviewers. We strongly encourage code deposition in a community repository (e.g. GitHub). See the Nature Portfolio guidelines for submitting code & software for further information.

## Data

Policy information about availability of data

All manuscripts must include a data availability statement. This statement should provide the following information, where applicable:

- Accession codes, unique identifiers, or web links for publicly available datasets
- A description of any restrictions on data availability
- For clinical datasets or third party data, please ensure that the statement adheres to our policy

The cryo-EM maps have been deposited into the Electron Microscopy Data Bank (accession numbers EMD-27991, EMD-28694 – 28719, and EMD-28722). Motion-corrected micrographs have been deposited into the EMPIAR database (accession number EMPIAR-11277). PDB 6TZ5 was used to set up molecular dynamics simulations. Source data from figures is included with the manuscript.

# Field-specific reporting

Please select the one below that is the best fit for your research. If you are not sure, read the appropriate sections before making your selection.

☒ Life sciences          ☐ Behavioural & social sciences          ☐ Ecological, evolutionary & environmental sciences

For a reference copy of the document with all sections, see nature.com/documents/nr-reporting-summary-flat.pdf

# Life sciences study design

All studies must disclose on these points even when the disclosure is negative.

| | |
|---|---|
| Sample size | Cryo-EM images were collected for each sample until there were sufficient sample views to perform 3D reconstructions at the desired resolutions. On the Krios, 1400-6000 micrographs were collected for each sample, resulting in 20,000 to 60,000 usable particles per sample and 2.8 to 4.3 Angstrom resolution reconstructions. On the Arctica, between 400 and 800 micrographs were collected for each sample, resulting in 1000 to 4000 usable particles per sample and 4 to 9.5 Angstrom resolution reconstructions. As the number of usable particles in each micrograph is highly variable due to differences in sample density, it was not possible to predetermine sample size. Simulations were run for 2.4 us after equilibration and were run at least 3 times independently. |
| Data exclusions | Particles from cryo-EM images were excluded from analysis if they were overlapping, damaged, nonuniform in filament diameter, or did not contain high resolution features. These exclusions were determined with 2D and 3D classification in RELION. |
| Replication | Biophysical analyses were repeated at least 3 times. Cryo-EM data were randomly divided into two halves that were independently refined. Simulations were repeated at least 3 times. All attempts at replicating the experimental and simulation data were successful. |
| Randomization | Data from each different sample is inherently randomized during cryo-EM data analysis. Particles are divided into random half sets and processed independently. Resolution estimates are based on comparison of independently processed half maps. |
| Blinding | Investigators were not blinded to sample identity, but were blinded with respect to the data randomization during cryo-EM data processing. All data sets from different samples were processed using the same parameters using a standardized data processing pipeline in RELION. Further blinding was not possible as CryoEM maps require a priori knowledge of the sample for interpretation. Investigators were not blinded to group allocation during other experiments and simulations as each sample was analyzed individually, not as part of a group. |

# Reporting for specific materials, systems and methods

We require information from authors about some types of materials, experimental systems and methods used in many studies. Here, indicate whether each material, system or method listed is relevant to your study. If you are not sure if a list item applies to your research, read the appropriate section before selecting a response.

## Materials & experimental systems

| n/a | Involved in the study |
|---|---|
| ☒ | ☐ Antibodies |
| ☒ | ☐ Eukaryotic cell lines |
| ☒ | ☐ Palaeontology and archaeology |
| ☒ | ☐ Animals and other organisms |
| ☒ | ☐ Human research participants |
| ☒ | ☐ Clinical data |
| ☒ | ☐ Dual use research of concern |

## Methods

| n/a | Involved in the study |
|---|---|
| ☒ | ☐ ChIP-seq |
| ☒ | ☐ Flow cytometry |
| ☒ | ☐ MRI-based neuroimaging |

