## [Peer Review File · Nature Structural & Molecular Biology]

Peer Review Information

Manuscript Title: Brominated Lipid Probes Expose Structural Asymmetries in Constricted Membranes

Corresponding author name(s): Adam Frost, Michael Grabe

Reviewer Comments & Decisions:

Decision Letter, initial version:
--

Message: 14th Mar 2022

Dear Adam,

Thank you again for submitting your manuscript "Brominated Lipid Probes Expose Structural Asymmetries in Constricted Membranes". I apologize for the delay in responding, which resulted from the difficulty in obtaining suitable referee reports. Nevertheless, we now have comments (below) from the 3 reviewers who evaluated your paper. In light of those reports, we remain interested in your study and would like to see your response to the comments of the referees, in the form of a revised manuscript.

You will see that while reviewers #1 and #2 are quite positive about the work and recommend publication after minor revisions, reviewer #3 is more critical. They suggest that the text is thoroughly revised to improve its clarity and placement of the results within the context of the literature. Furthermore, the referee advises to explain better the new insights into ESCRT function gleaned from the study. To this end, I provide some guidelines on our Article format below. Please be sure to address/respond to all concerns of the referees in full in a point-by-point response and highlight all changes in the revised manuscript text file. If you have comments that are intended for editors only, please include those in a separate cover letter.

Editorially, we feel that addressing the concerns of the reviewers to their full satisfaction will be important for us to consider the study further. However, we are committed to providing a fair and constructive peer-review process. Do not hesitate to contact us if there are specific requests from the reviewers that you believe are technically impossible or unlikely to yield a meaningful outcome.

We expect to see your revised manuscript within 6 weeks. If you cannot send it within this

time, please contact us to discuss an extension; we would still consider your revision, provided that no similar work has been accepted for publication at NSMB or published elsewhere.

To facilitate revising the manuscript, I am providing some guidelines for our Article format:

- abstract should be under 150 words, no references;
- main text is typically between 3,500 and 4,500 words, and should be organized as introduction, results (with subheadings <60 characters) and discussion;
- display items (figures and tables): typically between 6 and 8 (max.);
- max. 10 Extended Data Figures (embedded in HTML and PDF versions of the Article for quick accessibility);
- supplementary items: Up to 10 Supplementary Data Figures; other allowed supplementary items are Suppl Table, Note, Video, Data Set.
- uncropped images of gels and blots should be presented in a Supplementary Data Set.

Reporting Summary:

When submitting the revised version of your manuscript, please pay close attention to our [href="https://www.nature.com/nature-research/editorial-policies/image-integrity">Digital Image Integrity Guidelines.](https://www.nature.com/nature-research/editorial-policies/image-integrity)

SOURCE DATA: we urge authors to provide, in tabular form, the data underlying the graphical representations used in figures. This is to further increase transparency in data reporting, as detailed in this editorial (<http://www.nature.com/nsmb/journal/v22/n10/full/nsmb.3110.html>). Spreadsheets can be submitted in excel format. Only one (1) file per figure is permitted; thus, for multi-paneled figures, the source data for each panel should be clearly labeled in the Excel file; alternately the data can be provided as multiple, clearly labeled sheets in an Excel file. When submitting files, the title field should indicate which figure the source data pertains to. We encourage our authors to provide source data at the revision stage, so that they

are part of the peer-review process.

Data availability: this journal strongly supports public availability of data. All data used in accepted papers should be available via a public data repository, or alternatively, as Supplementary Information. If data can only be shared on request, please explain why in your Data Availability Statement, and also in the correspondence with your editor. Please note that for some data types, deposition in a public repository is mandatory - more information on our data deposition policies and available repositories can be found below: <https://www.nature.com/nature-research/editorial-policies/reporting-standards#availability-of-data>

[Redacted]

Kind regards,
Florian

Florian Ullrich, Ph.D.

Associate Editor
Nature Structural & Molecular Biology
ORCID 0000-0002-1153-2040

Referee expertise:

Referee #1: membrane deformation mechanisms

Referee #2: EM, membrane lipids, MD simulations

Referee #3: ESCRTS, structural biology, cryo-EM

Reviewers' Comments:

Reviewer #1:

Remarks to the Author:

In this manuscript, the Frost lab introduces the use of brominated lipids to increase the contrast of cryo-EM images at the level of the unsaturated bonds of lipids, most notably their polyunsaturated acyl chains. This technique allows the authors to get unprecedented information about the organisation of a lipid membrane underneath a membrane deforming machinery, here a model system consisting of human ESCRT-III proteins CHMP1B and IST1. They are, for the first time, in position to directly describe the local and global arrangements of lipids. They can distinguish the lipid conformation at different levels: between the concave and convex leaflets; between a flat and a curved membranes; between convex regions that are in direct contact with the ESCRTIII helices and convex regions that are between the ESCRTIII helices. This work is exceptional in term of novelty and quality. It is also impressive by the number of data generated, which include different lipid compositions different brominated lipids, as well as well chosen controls (the various mutants of F9 and F13). It is likely that the combination of Cryo-EM and halogenated lipids will be very useful in the future to better analyze the match between remodelling machineries and membranes of defined lipid compositions. I confess that I had difficulty in finding even a minor point, which would improve this outstanding study. I recommend publication.

Reviewer #2:

Remarks to the Author:

Highly fluid nature of biological membrane lipids are significant problems to determine their structures. By freezing the samples, snap shot of such highly fluidic structure can be imaged using cryo-TEM. However the images are of low contrast and not easy to be interpreted at high resolution.

Moss III et al. reported that brominated lipids are useful as contrast probes for cryo-EM to study nanoscale biological membrane structures. They employed highly focused ESCRT-III membrane remodeling system as a model target, and successfully assessed the composition and structure of the essential molecules of ESCRT-III system at a better resolution than the molecular level. By combining cryo-TEM with MD, they discussed the mechanism of ESCRT-III system to deform lipid membrane at atomic level. The text is well written, and both of the cryo-TEM images and negatively stained TEM images are

beautiful. The difference between brominated and unlabeled lipid membranes is clear in cryo-TEM reconstruction, and the results match to the results by coarse-grained-MD. Statistics were employed properly. Taken together with other papers, the importance of the proposed method seems clear especially for the structural biology related to lipid membranes. However, before recommending this paper for publication in the Nature Structural & Molecular Biology, the following points should be addressed.

Major

The combination between cryo-TEM and MD sounds reasonable because biological membrane lipid is usually highly fluid and not easy to reach molecular-level resolution using averaging for cryo-TEM. In simulations of high molecular weight objects using MD, they are generally influenced by the initial atomic coordinates of their molecules. However, the initial coordinates of the model and the description of why the authors take them reasonable are just simply written. The initial model including the atomic coordinates should be precisely described to reproduce the results.

Figure 1f: why is the cylinder is twisted while the reconstruction in Figure 1e is straight. It should be explained.

Minor

The idea of membrane-protein interaction-induced membrane structure deformation including tubular arrangement is also proposed for organelle and cell membrane by other scientists. It may be interesting to cite such papers and discuss applicability of this system to study such phenomena. For example, *Int. J. Mol. Sci.* 22, 2624 (2021).

References.

The following references should be corrected.

They used "PNAS" in reference 1 and 41, but "Proc Natl Acad Sci USA" in reference 20 and 21.

There are too many capitals in the titles in references 10, 15, 18, 26, 30, 32, 53, 54, 55 and 56.

Reviewer #3:

Remarks to the Author:

The study by Moss et al. delves into the lipid rearrangements needed for extreme ESCRT-coated membrane tube constriction previously seen by this group in their 2020 manuscript by Nguyen et al. I will leave the review of the computational part to the expert reviewer(s) in that domain. The main highlight of the cryo-EM part of the paper is the circled Br density seen in the zoomed panels of Fig. 3b, c. The authors used a 1982 method to make tail-labeled brominated lipids. The use of the Br label in cryo-EM is clever and reminds me

of the elegant series of papers from 1986 on by Stephen White using neutron diffraction to probe membrane structure. The additional signal from the Br atoms is clearly evident in the EM images. A large amount of cyro-EM data were collected and seem to have been analyzed appropriately.

The strength of the experimental part of this study is potential power of lipid tail labeling for cryo-EM studies of membranes. On the other hand, the application here is rather specialized. The main goal seems to be to understand how the unusual lipid SDPC enables a hyper-constricted state of the ESCRT system, whose physiological existence seems unlikely.

The manuscript needs improvement in its clarity and scholarship. The manuscript appears to have originally been drafted as a short report to Science, perhaps in haste, as judged by the cavalier referencing. The ms. suffers from a lack of clarity due to the compression of the text to fit this format. References 1-10 on membrane mechanics and 11-19 on cryo-EM structures of membrane proteins grouped together in an uninformative way. On pg. 10, line 18, too many references are bundled together again.

The authors make sweeping generalities about advancing structural biology as a whole, but they seemingly have little interest in the biology of the ESCRTs, the nominal topic of the manuscript.

The concluding paragraph (pg. 10, line 29) seems to propose that the work described in the manuscript will help research in AI-inferred structures, multistate inference, time-resolved studies, and in situ structural biology. The implications that this study, interesting as it is in some respects, will advance structural biology along these broad fronts, strikes me as an over-reach. I would have been more interested to hear what, if anything, was learned about ESCRT biology and mechanism.

One of the more interesting findings is the large compositional differences between the leaflets following equilibration of the MD simulations. So, are the authors proposing that CHMP1B and IST1 are phospholipid flippases? If so, what is the mechanism? Lipid flipping must have occurred during the course of the MD simulations. How did this happen?

Why the focus on Phe side-chains in the membrane at expense of other hydrophobic amino acids? Are there only Phe and no other hydrophobic side chains?

Fig. 1c,d were uninterpretable to me. This figure needs close up views, labels, plots of distances from tube center and protein.

In text, be clear which conclusions were drawn from MD and which from cryo-EM. Pg. 4, line 15, it needs to be clarified this conclusion is from MD.

Fig. 2 provide color code and use labels. Describing colors in the figure legend is insufficient given the complexity of the system.

Ref. 49, update for the appropriate journal.

Author Rebuttal to Initial comments

Response to Reviewers

Line-by-line responses in red.

Referee expertise:

Referee #1: membrane deformation mechanisms

Referee #2: EM, membrane lipids, MD simulations

Referee #3: ESCRTS, structural biology, cryo-EM

Reviewers' Comments:

Reviewer #1:

Remarks to the Author:

In this manuscript, the Frost lab introduces the use of brominated lipids to increase the contrast of cryo-EM images at the level of the unsaturated bonds of lipids, most notably their polyunsaturated acyl chains. This technique allows the authors to get unprecedented information about the organisation of a lipid membrane underneath a membrane deforming machinery, here a model system consisting of human ESCRT-III proteins CHMP1B and IST1. They are, for the first time, in position to directly describe the local and global arrangements of lipids. They can distinguish the lipid conformation at different levels: between the concave and convex leaflets; between a flat and a curved membranes; between convex regions that are in direct contact with the ESCRTIII helices and convex regions that are between the ESCRTIII helices. This work is exceptional in term of novelty and quality. It is also impressive by the number of data generated, which include different lipid compositions different brominated lipids, as well as well chosen controls (the various mutants of F9 and F13). It is likely that the combination of Cryo-EM and halogenated lipids will be very useful in the future to better analyze the match between remodelling machineries and membranes of defined lipid compositions. I confess that I had difficulty in finding even a minor point, which would improve this outstanding study. I recommend publication.

We thank the reviewer for their time and enthusiasm for our work.

Reviewer #2:

Remarks to the Author:

Highly fluid nature of biological membrane lipids are significant problems to determine their structures. By freezing the samples, snap shot of such highly fluidic structure can be imaged using cryo-TEM. However the images are of low contrast and not easy to be interpreted at high resolution.

Moss III et al. reported that brominated lipids are useful as contrast probes for cryo-EM to study nanoscale biological membrane structures. They employed highly focused ESCRT-III membrane remodeling system as a model target, and successfully assessed the composition and structure of the essential molecules of ESCRT-III system at a better resolution than the molecular level. By combining cryo-TEM with MD, they discussed the mechanism of ESCRT-III system to deform lipid membrane at atomic level. The text is well written, and both of the cryo-TEM images and negatively stained TEM images are beautiful. The difference between brominated and unlabeled lipid membranes is clear in cryo-TEM reconstruction, and the results match to the results by coarse-grained-MD. Statistics were employed properly. Taken together with other papers, the importance of the proposed method seems clear especially for the structural biology related to lipid membranes. However, before recommending this paper for publication in the Nature Structural & Molecular Biology, the following points should be addressed.

We thank the reviewer for their careful reading of the text and support for the manuscript.

Major

The combination between cryo-TEM and MD sounds reasonable because biological membrane lipid is usually highly fluid and not easy to reach molecular-level resolution using averaging for cryo-TEM. In simulations of high molecular weight objects using MD, they are generally influenced by the initial atomic coordinates of their molecules. However, the initial coordinates of the model and the description of why the authors take them reasonable are just simply written. The initial model including the atomic coordinates should be precisely described to reproduce the results.

We thank the reviewer for this helpful request. To assist other groups in reproducing our results, we have now added the set of scripts used to generate the initial states of protein and lipid packing used in our simulations, as well as one protein-only and one protein+lipid example initial coordinate file. Directions for their use have been added to the SI Section 12, and following review we will deposit the materials in Zenodo etc. We have also expanded on the descriptions of these procedures in the Methods (sections “Summary of Simulations” and “Sets 1 & 2”) and SI (Section 12). We have not included full sets of coordinates because, as described in the Methods and SI, the initial positions for lipids were randomized for each individual simulation replicate, followed by 10+ μ s of standard MD across several equilibration stages before the collection of production data. We note that despite the randomization of lipid coordinates, leaflet structure and lipid compositions (e.g., Table S7), as well as local lipid structures relative to the protein contact sites (e.g., ED Fig. 2f) were reproducible in each replicate. We believe these data demonstrate that our results are not governed by the initial conditions of the simulations.

Figure 1f: why is the cylinder is twisted while the reconstruction in Figure 1e is straight. It should be explained.

The two images come from different sources: the reconstruction in Figure 1e is from the cryo-EM experiments and is built from segments of protein-coated tubules that are 100s of nm in length. Coherent averaging of tens of thousands of filament segments and the application of helical symmetry result in a completely cylindrical bilayer tubule in the reconstruction. The image in Figure 1f is a snapshot from one of the CG-MD simulations, using a box of 18 nm in length that includes only two turns of the protein coat (Figure 1d). The imperfect periodicity of the simulated protein coat gives rise to boundary effects that influence the shape of the uncoated, exposed regions of the membrane tubule (Fig. 1d, Fig. S7, Fig. S8). This region fluctuates, free from the protein collar. We minimized this fluctuation by tuning the total amount of lipid used in simulation. We found that a total count of 1300 lipids produced the most cylindrical tubules while minimizing overpacking, although we still observe slight anisotropic bulges at the edge of the two-turn protein coat (see ED Fig. 1a-f). Inserted below is another example image from simulation that highlights the variability of bilayer shape outside the protein coat. We have changed Fig. 1f to this example as it is more representative of the replicates. We note that we do not use the regions outside of the protein coat for any of the analyses in this manuscript (see Fig. S7b). Additionally, we have added more text (page 3 line 18) explaining the shape of the membrane in Fig. 1e-f.

Minor

The idea of membrane-protein interaction-induced membrane structure deformation including tubular arrangement is also proposed for organelle and cell membrane by other scientists. It may be interesting to cite such papers and discuss applicability of this system to study such phenomena. For example, *Int. J. Mol. Sci.* 22, 2624 (2021).

We appreciate the suggestion and have added this reference and discussion (page 12 line 29) of the

potential applications of brominated lipid probes to many other examples of membrane shaping systems.

References.

The following references should be corrected.

They used "PNAS" in reference 1 and 41, but "Proc Natl Acad Sci USA" in reference 20 and 21.

These references have been fixed.

There are too many capitals in the titles in references 10, 15, 18, 26, 30, 32, 53, 54, 55 and 56.

These references have been fixed.

Reviewer #3:

Remarks to the Author:

The study by Moss et al. delves into the lipid rearrangements needed for extreme ESCRT-coated membrane tube constriction previously seen by this group in their 2020 manuscript by Nguyen et al. I will leave the review of the computational part to the expert reviewer(s) in that domain. The main highlight of the cryo-EM part of the paper is the circled Br density seen in the zoomed panels of Fig. 3b, c. The authors used a 1982 method to make tail-labeled brominated lipids. The use of the Br label in cryo-EM is clever and reminds me of the elegant series of papers from 1986 on by Stephen White using neutron diffraction to probe membrane structure. The additional signal from the Br atoms is clearly evident in the EM images. A large amount of cyro-EM data were collected and seem to have been analyzed appropriately.

The strength of the experimental part of this study is potential power of lipid tail labeling for cryo-EM studies of membranes. On the other hand, the application here is rather specialized. The main goal seems to be to understand how the unusual lipid SDPC enables a hyper-constricted state of the ESCRT system, whose physiological existence seems unlikely.

We thank the reviewer for their comments. However, we dispute the idea that the ESCRT system used in this manuscript is not physiologically relevant. While much work remains to be done in living cells, recent work by Cada et al. showed that ESCRT filaments constricted to close to the same diameter as ours are capable of catalyzing friction-driven membrane scission *in vitro*.¹ We agree that long filaments of membrane remodeling proteins like dynamins and ESCRTs likely do not exist *in vivo*, where a few turns of the filament likely catalyze fission.² However, with the exception of fission that proceeds via a

shearing mechanism,³ membrane fission must proceed via very highly constricted intermediate states, even if these states are only transient.^{3,4} Our work reveals the molecular basis for stabilizing such a highly curved membrane structure. We believe these results reveal parts of the energetics and potential routes for regulation of membrane remodeling. The fundamental geometrical changes required for membrane fusion and fission are independent of the particular protein complex employed, extending the impact of our results and methods beyond the scope of just the ESCRT-III proteins.

The manuscript needs improvement in its clarity and scholarship. The manuscript appears to have originally been drafted as a short report to Science, perhaps in haste, as judged by the cavalier referencing. The ms. suffers from a lack of clarity due to the compression of the text to fit this format. References 1-10 on membrane mechanics and 11-19 on cryo-EM structures of membrane proteins grouped together in an uninformative way. On pg. 10, line 18, too many references are bundled together again.

We have separated these references by topic. Additionally, we have added to the main text to improve its clarity.

The authors make sweeping generalities about advancing structural biology as a whole, but they seemingly have little interest in the biology of the ESCRTs, the nominal topic of the manuscript. The concluding paragraph (pg. 10, line 29) seems to propose that the work described in the manuscript will help research in AI-inferred structures, multistate inference, time-resolved studies, and in situ structural biology. The implications that this study, interesting as it is in some respects, will advance structural biology along these broad fronts, strikes me as an over-reach. I would have been more interested to hear what, if anything, was learned about ESCRT biology and mechanism.

We respectfully disagree that ESCRT biology is the main topic of this manuscript. Instead, we focused on developing new methods for measuring the structure of a fluid membrane. We used the ESCRT-III proteins CHMP1B and IST1 as a foundational model system because they hold bilayers in a uniformly constricted tubule with unprecedented membrane curvature. The membrane inside the CHMP1B/IST1 filament is very close to the point of spontaneous fission, providing the unique opportunity to study a fission intermediate that is normally only transient. The ESCRT filaments are highly ordered, allowing us to recover high resolution structural details for the bilayer using the protein as a high precision alignment fiducial. Additionally, the positioning of amphipathic helix 1 of CHMP1B at the bilayer surface provides a unique opportunity to study how this important and widespread protein motif interacts with membranes. Our results reach beyond the ESCRT's and inform our understanding of membrane shape changes in general. The main result relevant to ESCRT biology are that membrane remodeling by ESCRT-III proteins is highly dependent upon lipid composition, suggesting that cells can regulate mediated remodeling via local changes in lipid composition.

We have made our claims about far-reaching implications on page 12 more modest.

One of the more interesting findings is the large compositional differences between the leaflets following equilibration of the MD simulations. So, are the authors proposing that CHMP1B and IST1 are phospholipid flippases? If so, what is the mechanism? Lipid flipping must have occurred during the course of the MD simulations. How did this happen?

We agree that it is a striking and somewhat surprising finding that the protein coat stabilizes such a compositional difference across the leaflets, and we thank the reviewer for suggesting we improve our discussion of this observation. That said, we are not proposing that CHMP1B or IST1 are flippases.

First, we artificially allow lipids to flip during the equilibration stage of the simulations in an attempt to arrive at the most stable energetic membrane configuration. To do this, we adopt a procedure established by the developers of the Martini model for equilibrating curved membranes of mixed composition.⁵ During leaflet equilibration, we introduce a pair of hydrated pores in the tubule away from the protein at which lipids flip-flop during several microseconds of leaflet equilibration to reach consistent inner versus outer leaflet compositions between replicates, after which the pores are closed before beginning production. Please see page 18 line 34, SI Section 12, and ED Fig. 1g-h for detailed explanations of the simulation setup and equilibration.

Second, as stated above, we do not have any data that directly suggests the ESCRT-III coat can flip lipids. We do not observe phospholipid flipping during any stage of the simulations except for the equilibration (except for cholesterol which does flip-flop between bilayers on the sub microsecond timescale). The most straightforward explanation for the observed leaflet compositional differences in the experimental data is that the proteins concentrate lipids in the tubules according to their curvature preferences and electrostatic interactions. As shown in Extended Data Fig. 3a, many filaments retain uncoated vesicle “reservoirs” protruding from their tips. These uncoated parts of the bilayer may retain lipids that are underrepresented within the filaments. Additionally, we often observe uncoated vesicles and filaments that are not fully constricted, suggesting that there may be some heterogeneity in the lipid compositions of individual vesicles. We may be enriching for certain lipid compositions in our cryo-EM reconstructions by only averaging filaments with consistent diameters. We note that cholesterol can flip between leaflets on the time scale of our sample preparation.^{6,7} Also, the entire lumen of the filament is lined with cationic residues, creating large electric field gradients within the filament. This field gradient could catalyze the flipping of charged lipids between leaflets, but we have no experimental evidence for this effect at this time.

Why the focus on Phe side-chains in the membrane at expense of other hydrophobic amino acids? Are there only Phe and no other hydrophobic side chains?

Yes, F9 and F13 are the only hydrophobic residues facing the membrane. CHMP1B helix 1 also bends away from the membrane somewhat so that F9 and F13 have the only side chains that partially insert

into the bilayer. Finally, the only perturbations to membrane structure that we observe with both cryo-EM and MD simulations occur near F9 and F13. The charged residues on CHMP1B helix 1 are critical for membrane binding, and their mutation often results in a loss of membrane binding, but they do not appear to be directly involved in remodeling. We have improved our explanation of the focus on F9 and F13 on page 3 line 21 of the text.

Fig. 1c,d were uninterpretable to me. This figure needs close up views, labels, plots of distances from tube center and protein.

We have added more labels and distances to Figure 1c,d to clarify the involved components and their comparison to the cryo-EM images in Figure 1a,b. We added close-up views of the simulations to ED Fig. 1. For plots of distance between layers of the filaments, we refer the reader to Fig. 3d. For a more detailed view of the protein coat, we refer the reader to the PDB and our previous publication, Nguyen et al 2020, detailing this structure.⁸

In text, be clear which conclusions were drawn from MD and which from cryo-EM. Pg. 4, line 15, it needs to be clarified this conclusion is from MD.

We have clarified this point. We note that, with the exception of lipid diffusion rates and positions of unbrominated lipid tails, all of our conclusions in the paper are supported by both experiment and simulation.

Fig. 2 provide color code and use labels. Describing colors in the figure legend is insufficient given the complexity of the system.

We have added color-coded models of the lipids presented as well as labels to provide further clarity.

Ref. 49, update for the appropriate journal.

We have fixed this reference.

References

1. Cada, A. K. *et al.* *Reconstitution reveals friction-driven membrane scission by the human ESCRT-III proteins CHMP1B and IST1*. 2022.02.03.479062 <https://www.biorxiv.org/content/10.1101/2022.02.03.479062v2> (2022) doi:10.1101/2022.02.03.479062.

2. Adell, M. A. Y. *et al.* Recruitment dynamics of ESCRT-III and Vps4 to endosomes and implications for reverse membrane budding. *eLife* **6**, e31652 (2017).
3. Lenz, M., Morlot, S. & Roux, A. Mechanical requirements for membrane fission: common facts from various examples. *FEBS Lett* **583**, 3839–3846 (2009).
4. Kozlovsky, Y. & Kozlov, M. M. Membrane fission: model for intermediate structures. *Biophys J* **85**, 85–96 (2003).
5. Risselada, H. J. & Marrink, S. J. Curvature effects on lipid packing and dynamics in liposomes revealed by coarse grained molecular dynamics simulations. *Phys. Chem. Chem. Phys.* **11**, 2056–2067 (2009).
6. Bennett, W. F. D., MacCallum, J. L., Hinner, M. J., Marrink, S. J. & Tieleman, D. P. Molecular View of Cholesterol Flip-Flop and Chemical Potential in Different Membrane Environments. *J. Am. Chem. Soc.* **131**, 12714–12720 (2009).
7. Smith, R. J. M. & Green, C. The rate of cholesterol 'flip-flop' in lipid bilayers and its relation to membrane sterol pools. *FEBS Letters* **42**, 108–111 (1974).
8. Nguyen, H. C. *et al.* Membrane constriction and thinning by sequential ESCRT-III polymerization. *Nat Struct Mol Biol* **27**, 392–399 (2020).

Decision Letter, first revision:

Message:

Dear Dr. Frost,

Thank you for submitting your revised manuscript "Brominated Lipid Probes Expose Structural Asymmetries in Constricted Membranes" (NSMB-A45862A). Please accept my apologies for the unusual delay in reaching a decision. Your manuscript has now been seen by the original referees and their comments are below. The reviewers find that the paper has improved in revision, and therefore we'll be happy in principle to publish it in Nature Structural & Molecular Biology, pending minor revisions to comply with our editorial and formatting guidelines.

We are now performing detailed checks on your paper and will send you a checklist detailing our editorial and formatting requirements in about a week. Please do not upload

the final materials and make any revisions until you receive this additional information from us.

Sincerely,

Carolina

Carolina Perdigoto, PhD
Chief Editor
Nature Structural & Molecular Biology
orcid.org/0000-0002-5783-7106

Reviewer #2 (Remarks to the Author):

Since the authors addressed the reviewers' comments satisfactorily, I recommend this paper for publication in the Nature Structural and Molecular Biology.

Reviewer #3 (Remarks to the Author):

The authors have addressed all of my concerns.

Decision Letter, author guidance

Message: Our ref: NSMB-A45862A

18th Oct 2022

Dear Dr. Frost,

Thank you for your patience as we've prepared the guidelines for final submission of your Nature Structural & Molecular Biology manuscript, "Brominated Lipid Probes Expose Structural Asymmetries in Constricted Membranes" (NSMB-A45862A). Our sincerest apologies for the delay in providing you with the feedback needed to revise your manuscript. Please carefully follow the step-by-step instructions provided in the attached file, and add a response in each row of the table to indicate the changes that you have made. Please also check and comment on any additional marked-up edits we have proposed within the text. Ensuring that each point is addressed will help to ensure that your revised manuscript can be swiftly handed over to our production team.

We would like to start working on your revised paper, with all of the requested files and forms, as soon as possible. If you can resubmit within the next week it is possible that your submission could be published before the end of 2022. Please get in contact with us if you anticipate any delays in resubmission.

In recognition of the time and expertise our reviewers provide to Nature Structural & Molecular Biology's editorial process, we would like to formally acknowledge their contribution to the external peer review of your manuscript entitled "Brominated Lipid Probes Expose Structural Asymmetries in Constricted Membranes". For those reviewers who give their assent, we will be publishing their names alongside the published article.

Nature Structural & Molecular Biology offers a Transparent Peer Review option for new original research manuscripts submitted after December 1st, 2019. As part of this initiative, we encourage our authors to support increased transparency into the peer review process by agreeing to have the reviewer comments, author rebuttal letters, and editorial decision letters published as a Supplementary item. When you submit your final files please clearly state in your cover letter whether or not you would like to participate in this initiative. Please note that failure to state your preference will result in delays in accepting your manuscript for publication.

Cover suggestions

As you prepare your final files we encourage you to consider whether you have any images or illustrations that may be appropriate for use on the cover of Nature Structural & Molecular Biology.

Nature Structural & Molecular Biology has now transitioned to a unified Rights Collection system which will allow our Author Services team to quickly and easily collect the rights and permissions required to publish your work. Approximately 10 days after your paper is formally accepted, you will receive an email in providing you with a link to complete the grant of rights. If your paper is eligible for Open Access, our Author Services team will also be in touch regarding any additional information that may be required to arrange payment for your article.

Please note that *Nature Structural & Molecular Biology* is a Transformative Journal (TJ). Authors may publish their research with us through the traditional subscription access route or make their paper immediately open access through payment of an article-processing charge (APC). Authors will not be required to make a final decision about access to their article until it has been accepted. [Find out more about Transformative Journals](https://www.springernature.com/gp/open-research/transformative-journals)

Authors may need to take specific actions to achieve [compliance with funder and institutional open access mandates](https://www.springernature.com/gp/open-research/funding/policy-compliance-faqs). If your research is supported by a funder that requires immediate open access (e.g. according to [Plan S principles](https://www.springernature.com/gp/open-research/plan-s-compliance)) then you should select the gold OA route, and we will direct you to the compliant route where possible. For authors selecting the subscription publication route, the journal's standard licensing terms will need to be accepted, including [self-archiving policies](https://www.nature.com/nature-portfolio/editorial-policies/self-archiving-and-license-to-publish). Those licensing terms will supersede any other terms that the author or any third party may assert apply to any version of the manuscript.

Please use the following link for uploading these materials:
[Redacted]

Best regards,

Aimee Frier
Editorial Assistant
Nature Structural & Molecular Biology
nsmb@us.nature.com

On behalf of

Florian Ullrich, Ph.D.
Associate Editor
Nature Structural & Molecular Biology
ORCID 0000-0002-1153-2040

Reviewer #2:

Remarks to the Author:

Since the authors addressed the reviewers' comments satisfactorily, I recommend this paper for publication in the Nature Structural and Molecular Biology.

Reviewer #3:

Remarks to the Author:

The authors have addressed all of my concerns.

Final Decision Letter:

Message 11th Nov 2022

:

Dear Adam,

We are now happy to accept your revised paper "Brominated Lipid Probes Expose Structural Asymmetries in Constricted Membranes" for publication as a Article in Nature Structural & Molecular Biology.

As soon as your article is published, you can generate your shareable link by entering the DOI of your article here: http://authors.springernature.com/share. Corresponding authors will also receive an automated email with the shareable link

Your paper will be published online soon after we receive proof corrections and will appear in print in the next available issue. You can find out your date of online publication by contacting the production team shortly after sending your proof corrections. Content is published online weekly on Mondays and Thursdays, and the embargo is set at 16:00 London time (GMT)/11:00 am US Eastern time (EST) on the day of publication. Now is the time to inform your Public Relations or Press Office about your paper, as they might be interested in promoting its publication. This will allow them time to prepare an accurate and satisfactory press release. Include your manuscript tracking number (NSMB-A45862B) and our journal name, which they will need when they contact our press office.

About one week before your paper is published online, we shall be distributing a press release to news organizations worldwide, which may very well include details of your work. We are happy for your institution or funding agency to prepare its own press release, but it must mention the embargo date and Nature Structural & Molecular Biology. If you or your Press Office have any enquiries in the meantime, please contact press@nature.com.

Please note that *Nature Structural & Molecular Biology* is a Transformative Journal (TJ). Authors may publish their research with us through the traditional subscription access route or make their paper immediately open access through payment of an article-processing charge (APC). Authors will not be required to make a final decision about access to their article until it has been accepted. <https://www.springernature.com/gp/open-research/transformative-journals> Find out more about Transformative Journals

Kind regards,
Florian

Dr Florian Ullrich
Associate Editor, Nature
Consulting Editor, Nature Structural & Molecular Biology
ORCID 0000-0002-1153-2040
